# Deciphering intercellular signaling complexes by interaction-guided chemical proteomics

Jiangnan Zheng [1,4] ✉, Zhendong Zheng [1,2,4], Changying Fu[1,4], Yicheng Weng[1], An He[1], Xueting Ye[1], Weina Gao[1] & Ruijun Tian [1,3] ✉

Indirect cell–cell interactions mediated by secreted proteins and their plasma membrane receptors play essential roles for regulating intercellular signaling. However, systematic profiling of the interactions between living cell surface receptors and secretome from neighboring cells remains challenging. Here we develop a chemical proteomics approach, termed interaction-guided cross-linking (IGC), to identify ligand-receptor interactions in situ. By introducing glycan-based ligation and click chemistry, the IGC approach via glycan-to-glycan crosslinking successfully captures receptors from as few as 0.1 million living cells using only 10 ng of secreted ligand. The unparalleled sensitivity and selectivity allow systematic crosslinking and identification of ligand-receptor complexes formed between cell secretome and surfaceome in an unbiased and all-to-all manner, leading to the discovery of a ligand-receptor interaction between pancreatic cancer cell-secreted urokinase (PLAU) and neuropilin 1 (NRP1) on pancreatic cancer-associated fibroblasts. This approach is thus useful for systematic exploring new ligand-receptor pairs and discovering critical intercellular signaling events.

Cell–cell communication via secreted and plasma membrane proteins is essential for coordinating cellular activities such as cell proliferation, migration, and differentiation[1]. Cells in normal tissue sophistically control the secretion of signaling ligands (e.g., growth factors, cytokines, and chemokines) that bind to specific cell surface receptors, thereby maintaining cell homeostasis and normal tissue function. Dysregulation of physiological ligand-receptor interactions in tumor microenvironment is known to promote cancer growth and metastasis[2]. Therefore, unraveling the intercellular ligand-receptor interactions not only reveals fundamental biology, but also provides potential drug targets. Key to this effort is to develop robust methodology for systematically identifying these intercellular signaling complexes in biological contexts.

Current methods to profile human cell surface interactome can be broadly divided into ex situ and in situ methods. The ex situ methods, including classical biochemical screening techniques (e.g., Y2H) and affinity purification-mass spectrometry (AP-MS), can only reveal the biophysical interactions that may not occur in the biological microenvironment. Notably, membrane receptors typically contain hydrophobic transmembrane regions and extracellular glycosylation, rendering them difficult to study using methods that detect protein-protein interactions (PPIs) inside cells[3]. Moreover, AP-MS methods investigating PPIs in detergent-solubilized cell lysates tend to lose transient and weak ligand-receptor interactions[4], resulting in under-representation of extracellular interactions in current AP-MS datasets[5–7]. Conversely, in situ approaches are emerging to improve biological relevance. In situ crosslinking MS (XL-MS) is a promising solution to capture low-affinity interactions[8], while its identification sensitivity is still limited due to the low inter-protein crosslinking

[1]Department of Chemistry, School of Science, Southern University of Science and Technology, Shenzhen 518055, China. [2]School of Environment, Harbin Institute of Technology, Harbin 150090, China. [3]Research Center for Chemical Biology and Omics Analysis, School of Science, Southern University of Science and Technology, 1088 Xueyuan Road, Shenzhen 518055, China. [4]These authors contributed equally: Jiangnan Zheng, Zhendong Zheng, Changying Fu. ✉e-mail: zhengjn@sustech.edu.cn; tianrj@sustech.edu.cn

efficiency and sample complexity. Wollscheid and co-workers pioneered the selective ligand-receptor crosslinking on living cells by ligand-based receptor capture technology, including TRICEPS and HATRIC[9,10]. However, those methods typically require tens of micrograms of pure ligand and 20 million or more cells for each sample/replicate[11,12], making them difficult to study primary cells. Recently developed proximity labeling methods, such as PUP-IT[13], μMap[14], LUX-MS[15], and PhoTag[16], provide unique toolboxes for mapping cell surface PPIs of bait protein of interest. Nevertheless, those approaches are hypothesis-driven and best suited for studying the interactomes of the protein of interest.

Here, we present a hypothesis-free chemical proteomic strategy, termed interaction-guided crosslinking (IGC), to comprehensively unravel in situ intercellular signaling complexes on living cells (Fig. 1). Three trifunctional probes (Probe 1-3) with each possessing a ligand coupling group [N-hydroxysuccinimide (NHS) ester or aminooxy group], a crosslinking group (diazirine or alkyne group) and a biotin group were designed and employed for IGC (Supplementary Fig. 1–6). The spacer arm length of these probes was designed to be approximately 60 Å, which is well suited for inter-protein crosslinking[17]. The secretome in cell-conditioned medium (CM) were first conjugated to probe via NHS ester chemistry or oxime ligation after mild oxidation of glycans to generate aldehyde groups. The secretome-probe conjugates then bound to the living cell surface receptors via specific ligand-receptor interactions, while the other unbound proteins were washed away. Thus, the ligand-receptor complexes can be selectively crosslinked in situ under physiological conditions upon UV irradiation or Cu(I) catalyst addition. The crosslinked complexes were enriched by streptavidin beads, enzymatically digested and identified through LC–MS/MS analysis. Effective ligand coupling and receptor crosslinking are essential for successful capture of ligand-receptor complexes at the endogenous level. We adopted the glycan-based ligand conjugation to ensure the efficient labeling of low-abundant glycosylated ligands among the entire set of secreted proteins. In addition, the use of metabolic glycan labeling and cell-compatible click chemistry allows highly efficient crosslinking[18,19]. The IGC method with high sensitivity and specificity allows to systematically reveal indirect cell–cell interactions between pancreatic cancer-associated fibroblast cells (CAFs) and pancreatic cancer cells, and its value were exemplified by the discovery of a novel ligand-receptor interaction.

## Results

### Development of the Photo-IGC approach as a highly specific and universal method for receptor identification

The alkyl diazirine-based photoaffinity probes have been extensively exploited for the study of PPIs under native conditions[20]. Owing to the fast reaction kinetics ($10^7 – 10^9$ M$^{-1}$ s$^{-1}$) upon photoactivation and short lifetimes of the carbene intermediates (half-life in nanoseconds)[21], we reasoned that the diazirine-based photoreactive interaction-guided crosslinking (Photo-IGC) could provide great crosslinking specificity. In addition, the carbene intermediates can insert into almost any type of amino acid residues or glycans of the proximal proteins, thus eliminating the need for chemical labeling or genetic engineering of the cells. We first test the feasibility and versatility of Photo-IGC for identifying cell surface receptors by using well-known ligands, including epidermal growth factor (EGF), hepatocyte growth factor (HGF), insulin (INS), and platelet-derived growth factor B (PDGF-B). EGF and control samples were labeled with probe 1 and subjected to Photo-IGC experiments on 1 million HeLa cells, respectively (Fig. 2a). Both the incubation and crosslinking steps were performed in PBS buffer at 4 °C to prevent receptor internalization. Label-free quantification (LFQ) comparison between two groups successfully revealed epidermal growth factor receptor (EGFR) as EGF receptor (Fig. 2b). Moreover, EGFR can be confidently identified from only 0.2 million HeLa cells using as low as 10 ng of EGF per replicate (Supplementary Fig. 7). Likewise, Photo-IGC was able to identify hepatocyte growth factor receptor (MET), insulin receptor (INSR) and insulin receptor insulin-like growth factor I receptor (IGF1R) from HeLa cells (Fig. 2c), which are known receptors of HGF and INS, respectively. Molecules that do not share a receptor with the ligand can be used as negative controls. For instance, Photo-IGC experiments using glycine, Tris, BSA, or EGF as negative controls also successfully identified MET as the HGF receptor (Supplementary Fig. 8a–g). The PDGF receptors (Pdgfra and Pdgfrb) on mouse fibroblast NIH 3T3 cells were also identified using PDGF-B as the ligand. Importantly, almost none of the irrelevant proteins were co-purified in these results, demonstrating the high selectivity of Photo-IGC approach. Furthermore, we applied Photo-IGC to identify the receptor of SARS-CoV-2 by using recombinant SARS-CoV-2 spike receptor-binding domain (RBD) as ligand, and found its reported entry receptor ACE2 on monkey kidney Vero E6 cells.

NHS ester labeling is considered a universal protein conjugation method since proteins typically contain at least one free amino group, such as a free N-terminus or lysine residue. Theoretically, the

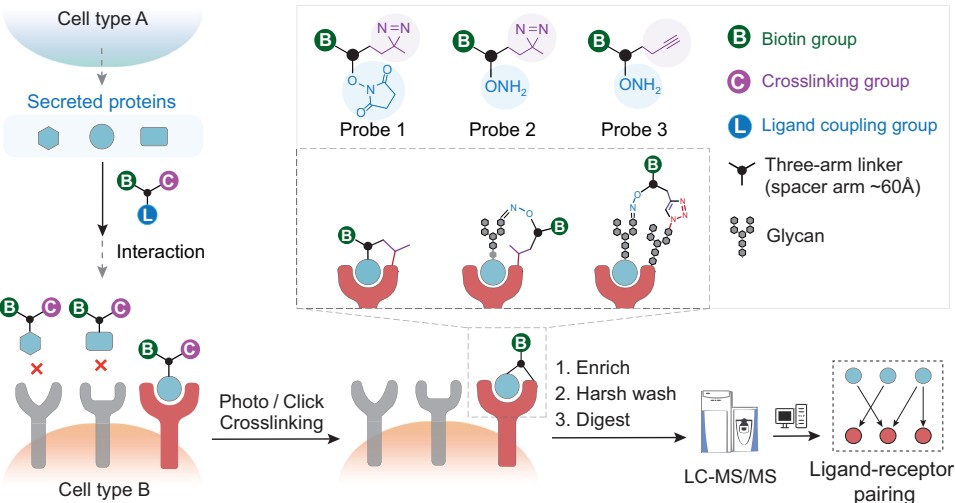

**Fig. 1 | Schematic of interaction-guided crosslinking (IGC) approach for ligand-receptor pair identification on living cells.** The IGC workflow is performed as follows: (1) the secreted proteins in conditioned media were labeled with trifunctional probes and bound to their receptors via specific ligand-receptor interactions. (2) The ligands and receptors were selectively crosslinked through UV irradiation or addition of the click reaction catalyst. (3) The biotinylated ligand-receptor complexes were enriched, digested, and identified by LC-MS/MS analysis.

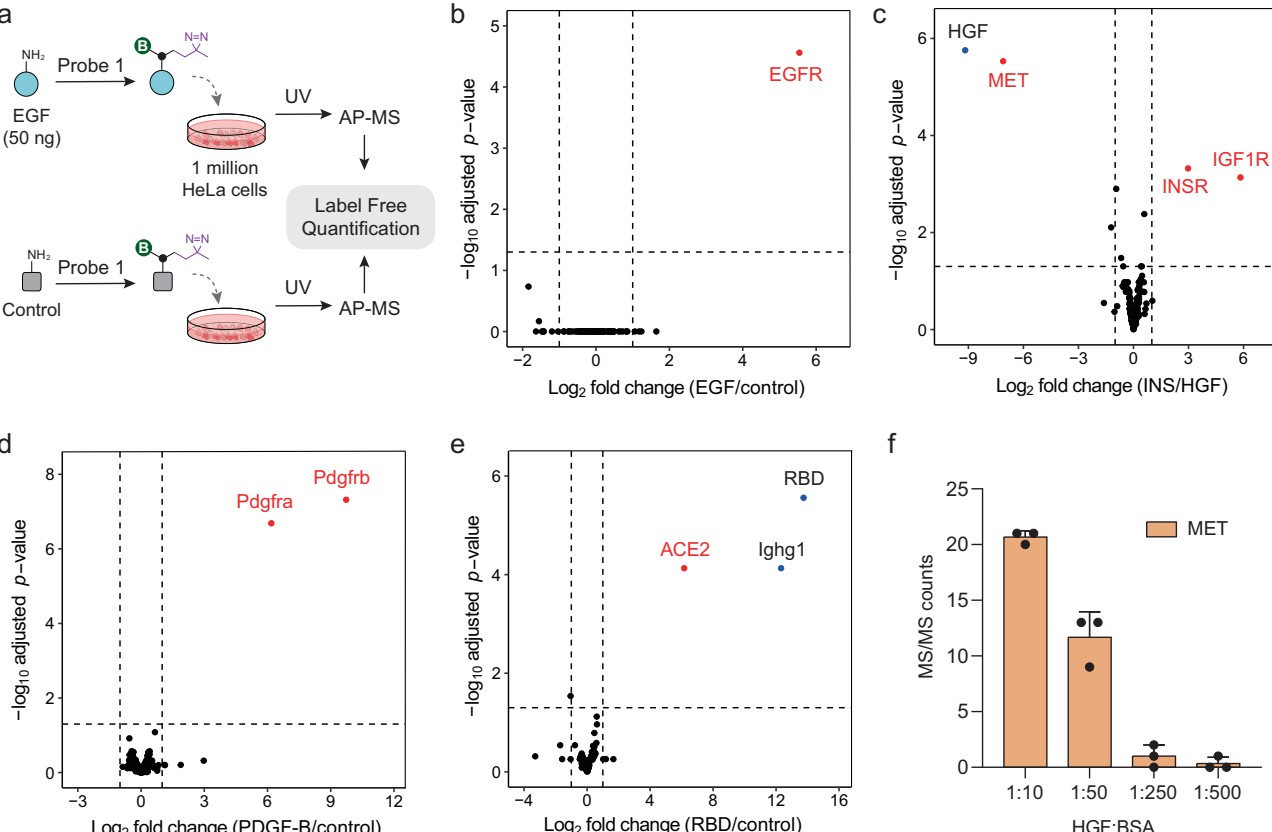

**Fig. 2 | Identification of receptors of purified ligand by Photo-IGC approach.**
**a** The ligand of interest or a control ligand is first coupled to the probe 1 and then incubated with cells. After 5 min UV light irradiation, cells were subjected to AP–MS analysis. The LFQ comparison between the ligand and control groups should reveal the receptors. **b** Photo-IGC with 50 ng of EGF on 1 million HeLa cells. Glycine was used as a negative control. **c** Photo-IGC experiments with 500 ng of porcine insulin (INS) and HGF as ligands were performed on 6 million HeLa cells. **d** Photo-IGC with 100 ng of PDGF-B on 2 million NIH 3T3 cells. Tris was used as a negative control.

**e** Photo-IGC with 1.6 µg of recombinant SARS-CoV-2 RBD-mFc on 6 million Vero E6 cells. BSA was used as a negative control. The known target receptors are highlighted in red. **f** Performance of Photo-IGC (probe 1) for the low-abundant ligand. HGF was mixed with different amount of BSA and labeled with probe 1, and 60 ng of labeled was used to capture receptor on 2 million HeLa cells. Data are presented as mean ± SD ($n$ = 3 biological replicates). All experiments were performed in triplicates per condition. Source data are provided as a Source Data file.

conjugation condition should be adjusted for each protein to avoid under- or over-labeling. We determined the optimal ligand-to-probe ratio for probe 1 labeling using two ligands with different molecular sizes, HGF (79.4 kDa with 47 lysine residues) and EGF (6.2 kDa with 2 lysine residues) as model proteins and the MS/MS count of identified receptors as a metric. A ligand-to-probe mass ratio of 1:2 was shown to be optimal for both HGF and EGF labeling (Supplementary Fig. 8h, i). This rule of thumb was successfully applied to other ligands for Photo-IGC (Fig. 2c, d). As the enrichment selectivity is critical to achieve high sensitivity of IGC method, we optimized the amount of streptavidin beads to reduce non-specific binding proteins, and found that 1 µL of streptavidin beads per 1 million cells is optimal to capture biotinylated proteins (Supplementary Fig. 8j, k).

To further test the labeling performance of probe 1 for low-abundant ligands, we added different amounts of bovine serum albumin (BSA) to the HGF solution and used them for the Photo-IGC experiments. As the amount of BSA increased, the amount of identified MET decreased (Fig. 2f). MET was hardly quantified when the HGF:BSA ratio reached 1:250, suggesting a limitation of NHS ester chemistry for labeling low-abundant ligands.

## Development of the Click-IGC approach for identifying low-abundant ligand-receptor complexes

Since most secreted and cell surface proteins are glycosylated, we sought to conjugate low-abundance ligands through oxime ligation

(probe 2 and 3). In contrast to the non-selective labeling of lysine residues that are often found at protein binding interfaces[22], glycan-based ligation should avoid the disruption of protein-protein interactions. On the other hand, to improve crosslinking efficiency, we also exploited the rapid copper(I) catalyzed azide-alkyne click chemistry (CuAAC) for click-crosslinking (termed Click-IGC). Accordingly, the glycans of cell surface glycoproteins were incorporated with azide groups by metabolic labeling with the well-studied sugar analogs[23], including Ac$_4$ManNAz, Ac$_4$GalNAz, and Ac$_4$GlcNAz. The sialic acid analog Ac$_4$ManNAz showed the best efficiency for labeling cell surface glycoproteins on HeLa cells and was therefore chosen for the following experiments (Fig. 3b). We then tested the CuAAC condition on the Ac$_4$ManNAz labeled K562 cells, and found that 50 µM of Cu(I) catalyst in PBS was sufficient to catalyze the labeling of cells with probe 3 in 15 min at 4 °C (Fig. 3c), which was consistent with previous reports[18,19]. We then used HGF-BSA mixture to test the performance of Click-IGC. As expected, MET can be identified from 1 million HeLa cells or NIH 3T3 with HGF-BSA (1:250 or 1:500) mixtures as ligands (Fig. 3d and Supplementary Fig. 9). More surprising, Click-IGC enabled the confident identification of MET on as little as 0.1 million HeLa cells using merely 10 ng of HGF mixed with 1000-fold of BSA (Fig. 3e), indicating the high sensitivity and advantage on discovering receptors of low-abundance ligands.

To better understand the differences of three probes in ligand conjugation and receptor crosslinking, we compared the performance

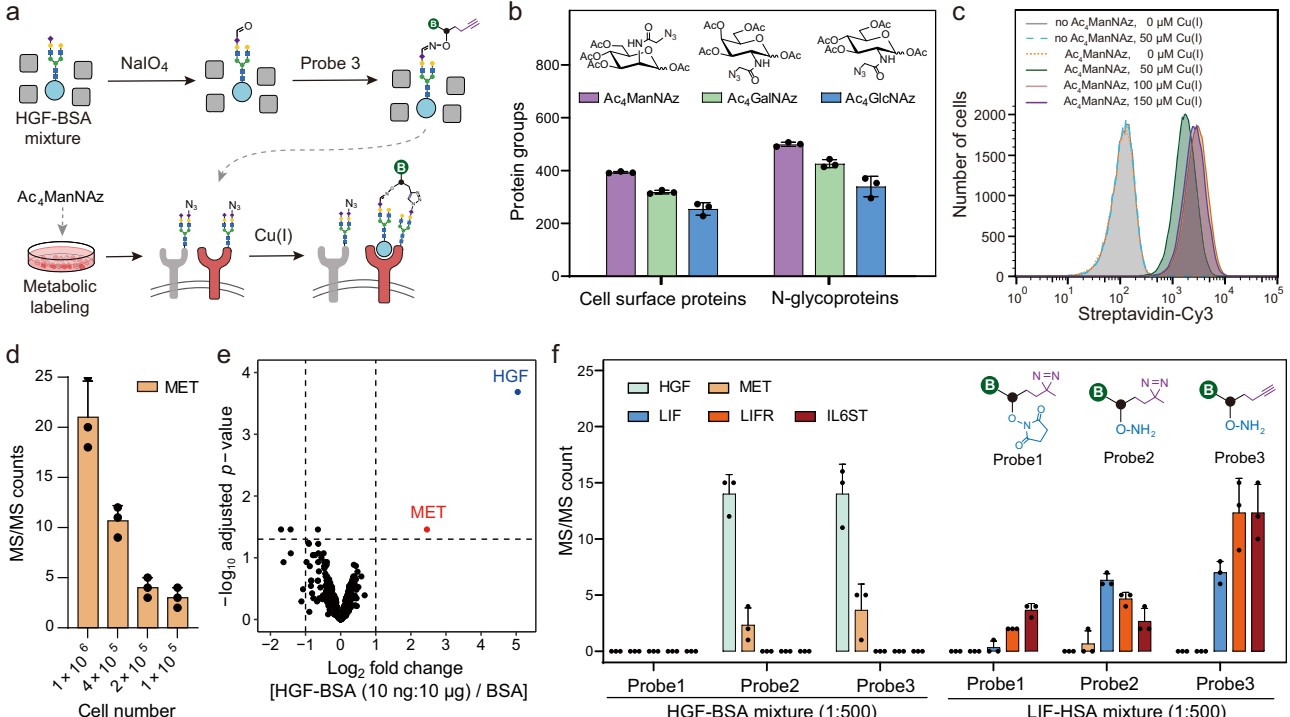

**Fig. 3 | Development of Click-IGC for identification of receptors of ligand mixture. a** The secretome is first coupled to the probe 3 and then incubated with Ac₄ManNAz labeled cells. After adding Cu(I) as catalyst of CuAAC click reaction for 15 min, cells were subjected to AP−MS analysis. **b** Metabolic labeling of cell surface proteins using Ac₄ManNAz, Ac₄GalNAz and Ac₄GlcNAz. Data are presented as mean ± SD (*n* = 3 biological replicates). **c** Flow cytometric analysis of Ac₄ManNAz labeled and unlabeled K562 cells incubated with probe 3 in the absence or presence of different concentration of Cu(I), and then conjugated with Streptavidin-Cy3.

**d** Identification of MET by Click-IGC using HGF-BSA mixture (HGF:BSA = 1:250) as ligand on 1, 0.4, and 0.2 million HeLa cells, respectively. Data are presented as mean ± SD (*n* = 3 biological replicates). **e** Click-IGC with HGF-BSA solution (HGF:BSA = 1:1000) as ligand on 0.1 million HeLa cells. The MS/MS count of MET is shown in (**d**). **f** Photo-IGC (probe 1 or probe 2) or Click-IGC (probe 3) with HGF-BSA (30 ng:15 μg) or LIF-HSA (30 ng:15 μg) on 1 million HeLa cells. Data are presented as mean ± SD (*n* = 3 biological replicates). Source data are provided as a Source Data file.

of three probes for identifying the receptors of low-abundance ligands using HGF-BSA and leukemia inhibitory factor (LIF)-human serum albumin (HSA) mixtures as model ligands. Since the key diffidence between probe 1 and probe 2 is the labeling group, the identification of HGF and LIF in probe 2 group but not in probe 1 group suggested that the low-abundance ligands can be efficiently labeled by oxime ligation but not by NHS labeling in the present of high-abundance non-glycosylated proteins. (Fig. 3f). Moreover, with the same ligand labeling group, probe 3 has the same ability to label and identify HGF and LIF. Accordingly, their corresponding receptors, i.e., MET, LIF receptor (LIFR) and co-receptor IL6ST were also successfully quantified by probe 2/3-based IGC (Fig. 3f). Compared with probe 2-based Photo-IGC, Click-IGC shown even better MET and LIFR identification, indicating the crosslinking efficiency of CuAAC reaction is higher than diazirine-based photocrosslinking, especially for glycosylated proteins. Overall, Click-IGC is a highly sensitive approach to identify receptors of low-abundant glycosylated ligands.

## Unbiased profiling of intercellular signaling complexes in all-to-all mode

To test the effectiveness of IGC strategies for discovering functional intercellular ligand-receptor pairs in paracrine signaling, we set up a co-culture model system using readily available cancer cell lines (Fig. 4a). KP4 pancreatic cancer cell line was chosen as the signaling sender because it is known to express a high level of HGF and a low level of LIF, which was confirmed by analysis of the CM of KP4 cells (Fig. 4b). As shown in the surfaceome analysis, HeLa cells can serve as the signaling receivers due to their expression of MET, LIFR, IL6ST, and many other well-known receptors (Fig. 4c). Using KP4-CM as ligands, all three IGC methods successfully captured MET and several other

receptors on HeLa cells (Fig. 4d and Supplementary Fig. 10). The Gene Ontology (GO) analysis of the significantly changed proteins revealed that both "cell surface" and "plasma membrane" were among the top 5 most enriched cellular component in both probe 2 and probe 3 groups (Fig. 4e and Supplementary Fig. 11). Meanwhile, probe 2 and probe 3 groups had fewer background binding (such as cytosol proteins) than probe 1 group and thus had higher percentages of cell surface/plasma membrane proteins (Fig. 4f). After pairing these putative receptors with KP4-secreted proteins, Click-IGC was found to identify the largest number of reported ligand-receptor pairs compared to Photo-IGC (Fig. 4f).

To further evaluate the interaction potency of the paired receptors, we generated the interaction score that reflects changes in relative protein abundance before and after pull-down, using surfaceome data as a reference (Fig. 4g). In accordance with high expression level of HGF in CM, MET was ranked as one of the most strongly interacting receptors in both Photo-IGC and Click-IGC. Notably, only Click-IGC can identify LIFR despite the low-abundance of its ligand LIF in CM. We then sought to determine whether such low level of LIF play a role in paracrine signaling. Since the formation of LIF-LIFR-IL6ST complex activates the JAK-STAT3 pathway[24], the phosphorylated STAT3 were detected by western blot to confirm the stimulation of HeLa cells. Incubation of HeLa cells with unlabeled or probe labeled KP4-CM resulted in STAT3 activation (Fig. 4h). After adding anti-LIF monoclonal antibody to neutralize LIF, the resulting probe 3 labeled CM cannot activate STAT3, indicating that LIF is a functional paracrine factor. Overall, IGC is a promising approach for all-to-all profiling functional intercellular signaling complexes in paracrine signaling, and Click-IGC should be the method of choice when the ligands are in low-abundance.

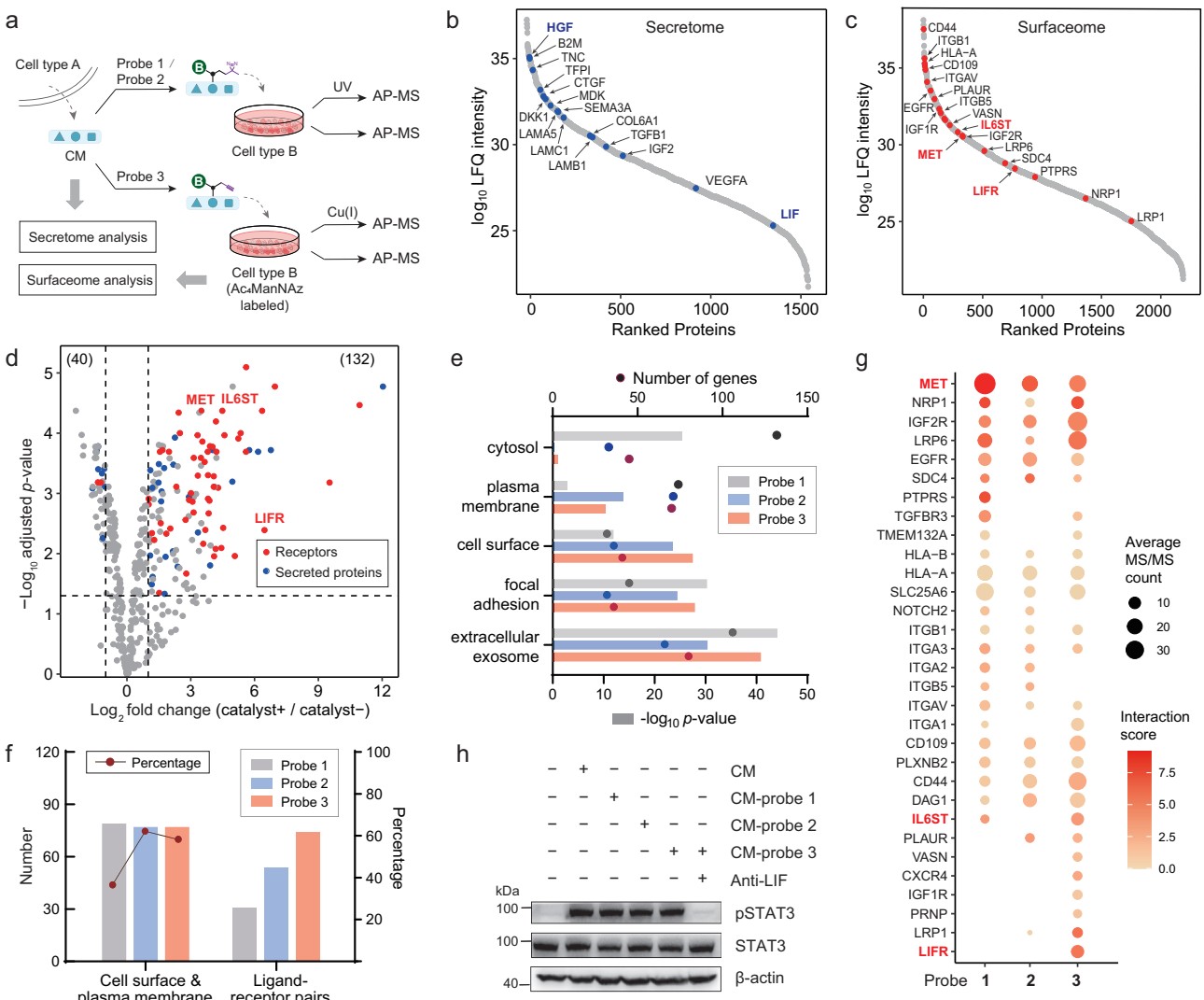

**Fig. 4 | Identification of receptors in paracrine signaling by Click-IGC and Photo-IGC approaches. a** The secretome in conditioned media (CM) of KP4 cells were conjugated with probe 1, probe 2, and probe 3, respectively, and used to capture their receptors on HeLa cells. All experiments were performed in triplicate per condition. **b** Abundance of secreted proteins in KP4-CM. **c** Surfaceome analysis of Ac₄ManNAz labeled HeLa cells using CuAAC-based biotinylation. **d** Click-IGC with KP4-CM as ligands on HeLa cells. The UniProt annotated receptors and secreted proteins are highlighted in red and blue, respectively. **e** Gene Ontology annotations for cell localization of the significant changed proteins in Click-IGC and Photo-IGC experiments (Fig. 4d and Supplementary Fig. 10). **f** The significant changed cell surface and plasma membrane proteins in Fig. 4e and the corresponding potential ligand-receptor pairs. **g** The annotated cell surface receptors identified by different IGC methods. **h** Western blot validation of LIF as the signaling molecule responsible for the activation of STAT3 in HeLa cells. Images are representative of 3 biological replicates. Source data are provided as a Source Data file.

## Deciphering the paracrine communication between pancreatic cancer cells and CAFs

Finally, we challenged the Click-IGC approach by studying bidirectional intercellular communication involving pancreatic cancer cells (PCCs) and pancreatic CAFs, which are limited in availability. In pancreatic tumor microenvironment, pancreatic stellate cells (PSCs) are the predominant CAFs and have a reciprocal relationship with PCCs[25,26]. To systematically explore the intercellular ligand-receptor pairs in the paracrine communication between PSCs and PCCs, we performed integrated quantitative proteomic analyses, combining the Click-IGC-based ligand-receptor pair analysis with secretome and surfaceome profiling (Fig. 5a). Two commonly used pancreas ductal adenocarcinoma cell lines, MIA PaCa-2 and PANC-1, were used as representative PCCs to interact with PSCs (using as few as 1 million cells per replicate). To analyze PCC-to-PSC cell communication, the CM of PCCs were first labeled with probe 3 containing a clickable group (termed cCM) and incubated with PSCs metabolically labeled

with Ac₄ManNAz. The interacting secreted proteins were revealed by quantitative proteomic comparison of the cCM-treated and -untreated PSCs (Fig. 5b and Supplementary Fig. 12).

From both MIA PaCa-2-CM and PANC-1-CM, we identified 36 secreted proteins that were captured onto PSC cell surface. Importantly, from the Click-IGC experiments using MIA PaCa-2-cCM and PANC-1-cCM, 6 putative receptors were commonly identified on PSCs including the CAFs markers, PDGFRA and PDGFRB[27,28] (Fig. 5c and Supplementary Fig. 12). Next, the PSC-to-PCC paracrine analysis revealed 40 putative ligands and 6 putative receptors using the same strategy (Fig. 5d–f and Supplementary Fig. 13). To obtain the interaction scores of the interacting proteins, we carried out secretome and surfaceome analysis of PCCs and PSCs, respectively (Fig. 5a, d, Supplementary Fig. 14). Finally, the bidirectional interactions between PCCs and PSCs were inferred by pairing the identified putative ligands and receptors (Fig. 5f, g). In general, limited number of receptors were identified on both sides, while a large number of

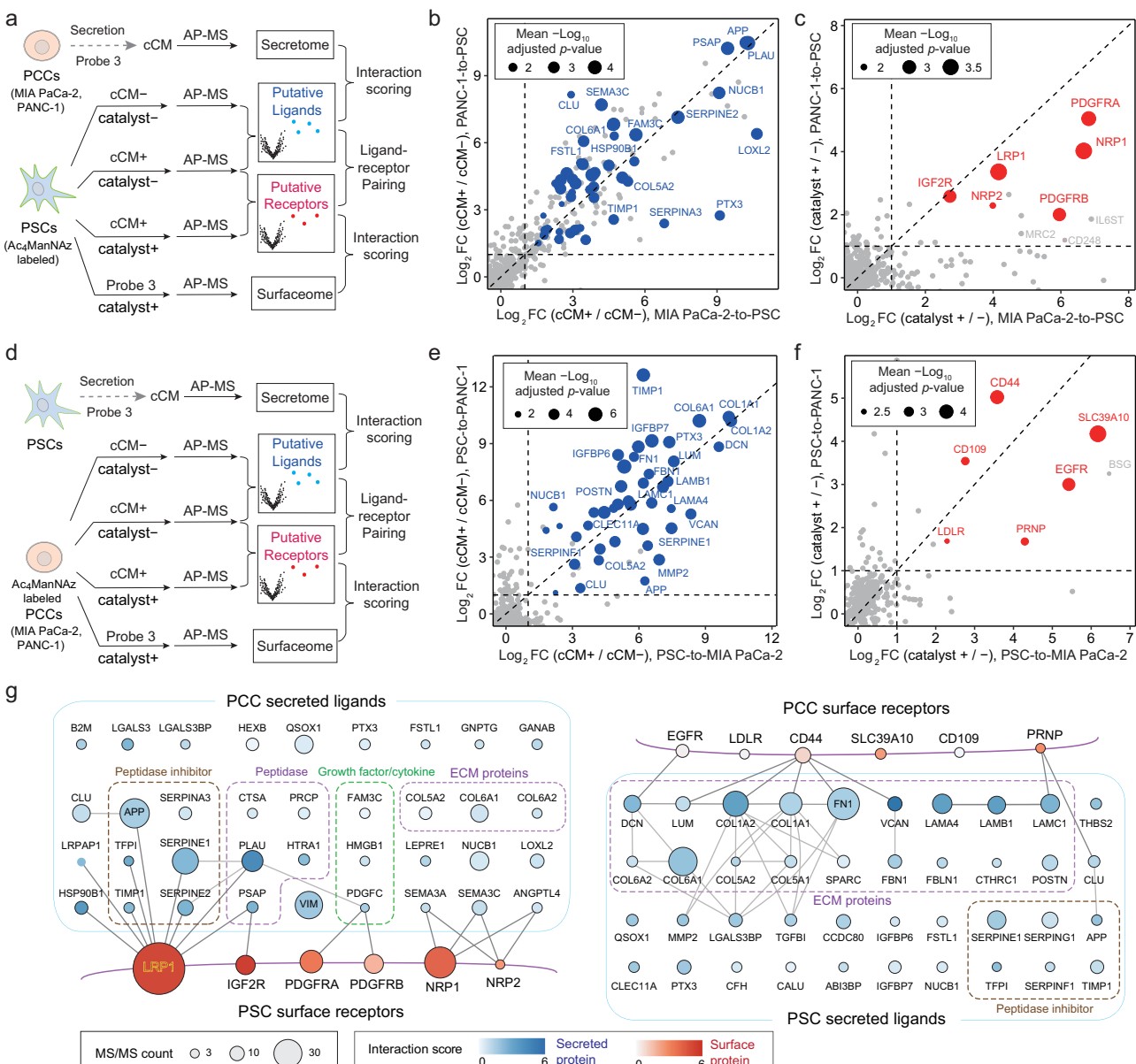

**Fig. 5 | Deciphering paracrine signaling between PSCs and PCCs. a** workflow for quantitative analysis of putative ligands and receptors in PCC-to-PSC paracrine signaling. The CM of PCCs were labeled with probe 3 (called PCC-cCM) for the AP–MS analysis of the PCC secretome. The AP–MS analysis of PCC-cCM-treated and untreated PSCs will reveal the putative ligands interacting with PSCs. The interaction possibilities of those ligands were scored using their relative protein abundance changes in pull-down samples and in CM. The Click-IGC on Ac₄ManNAz labeled PSCs using PCC-cCM will reveal the putative receptors. The Ac₄ManNAz labeled PSCs were also labeled with probe 3 and subjected to AP–MS analysis. The obtained PSC surfaceome were used to evaluate the interaction possibility of the putative receptors. **b** Comparison of the MIA PaCa-2 and PANC-1 secreted proteins that were interacted with PSCs. Only proteins quantified in both cell lines were shown, and the significant proteins annotated as "secreted" in UniProt were highlighted in blue. Dot *size* of those secreted proteins represents the mean $-\log_{10}$

adjusted $p$-values. **c** Comparison of the interacting surface proteins on PSC cells using MIA PaCa-2-cCM and PANC-1-cCM. The significant proteins annotated as plasma membrane in UniProt were highlighted in red. The size of each dot is proportional to the average $-\log_{10}$ adjusted $p$-value. **d** workflow for quantitative analysis of putative ligands and receptors in PSC-to-PCC paracrine signaling. **e** Comparison of secreted proteins interacting with PSC cells in MIA PaCa-2-cCM and PANC-1-cCM. **f** Comparison of the interacting surface proteins on MIA PaCa-2 and PANC-1 cells using PSC-CM. **g** Protein–protein interaction network analysis of the significant surface and secreted proteins in PCC-to-PSC (data in Fig. 5b and c) and PSC-to-PCC (data in Fig. 5d and e) paracrine signaling. The *lines* indicate the reported interactions of the identified cell surface proteins. Dot *size* represents the mean MS/MS count between two PCC cell types. Dot *color* represents the mean interaction score. All experiments were performed in triplicate. Source data are provided as a Source Data file.

secreted ligands with diverse functionalities were identified from PCC. Urokinase-type plasminogen activator (PLAU), a serine protease involved in cancer metastasis[29], was identified as the most significantly changed protein (Fig. 2b) and the highest scoring ligand in PCC-to-PSC paracrine signaling (Fig. 5g). Among the ligands with high interaction score and MS/MS count, we also identified the inhibitor of PLAU, plasminogen activator inhibitor-1 (SERPINE1),

which has been reported to be secreted by PCCs and can active PSCs through its receptor, the LDL receptor-related protein 1 (LRP1)[30]. On the other hand, the highest scoring receptor, LRP1, was reported to interact with PLAU either individually or in complex with SERPINE1[31,32]. NRP1 is a non-tyrosine kinase receptor which is highly expressed in pancreatic cancer[33] and function as a co-receptor for multiple signaling ligands, such as class 3 semaphorins (SEMA3A and

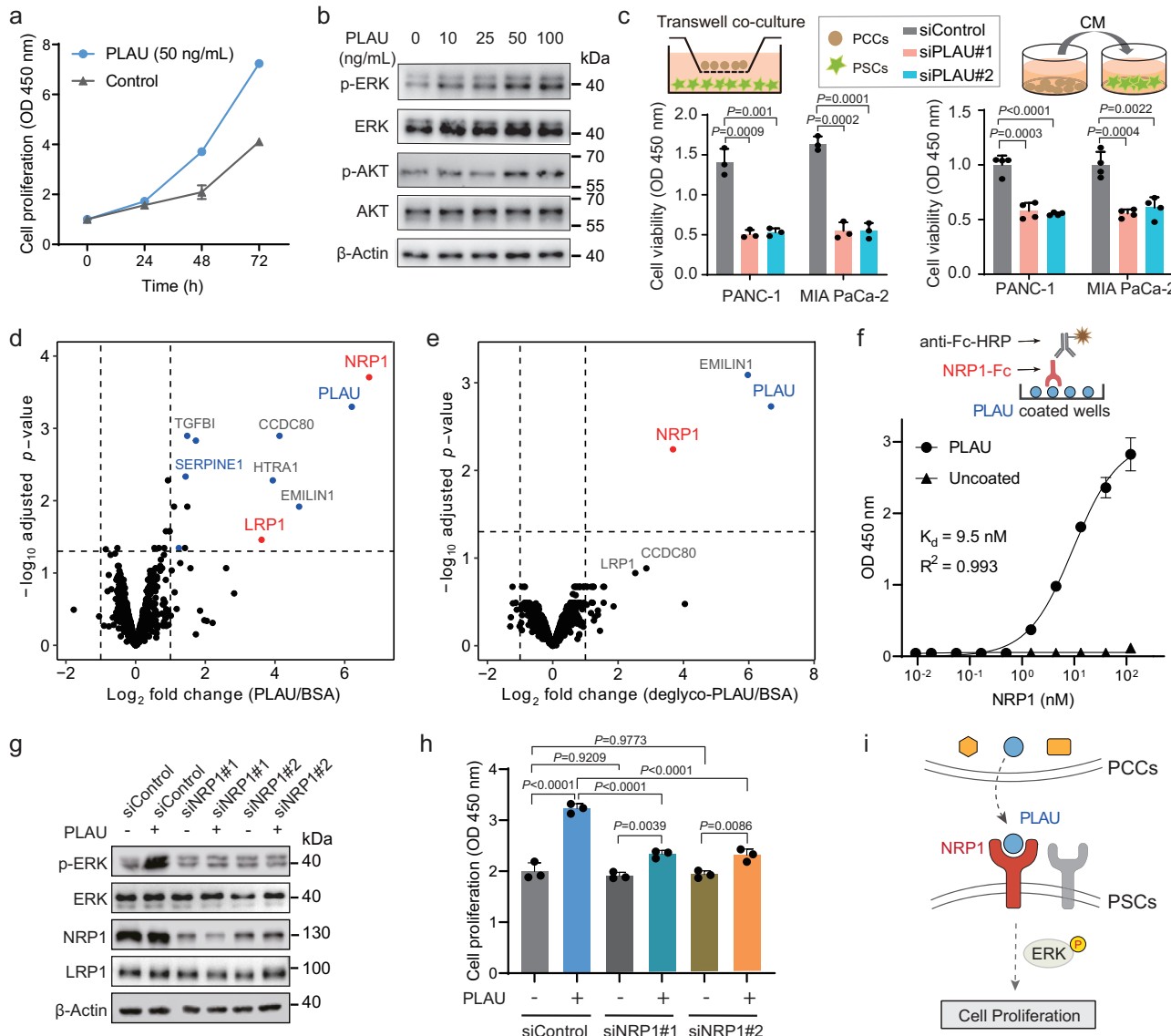

**Fig. 6 | Discovering PLAU and its novel receptor NRP1 as functional signaling molecules in PCC-to-PSC paracrine signaling.** Data are presented as the mean ± SD ($n$ = 4 biological replicates). **a** CCK-8 analysis of the effects of PLAU treatment on PSC cell growth. **b** Western blot analysis showing dose-dependent (10–100 ng/mL) activation of PLAU to PSC cells with the indicated antibodies. Images are representative of 3 biological replicates. **c** Cell viability of PSC cells co-cultured with PCCs ($n$ = 3 biological replicates) or cultured with CM of PCCs ($n$ = 4 biological replicates) which were transfected with negative control siRNA (siControl) and PLAU-siRNA. The $P$ values were calculated using two-sided Student's $t$-test and data are presented as the mean ± SD. **d, e** Photo-IGC experiments with **d** PLAU or **e** PNGase F treated PLAU as ligands were performed on PSC cells. BSA was used as a negative control. Significant proteins annotated as secreted and plasma membrane proteins were highlighted in blue and red, respectively. All IGC experiments were performed

in triplicate. **f** The affinity constant of the interaction between PLAU and NRP1. Microtiter plates were coated with 1.5 ng/μL PLAU or left uncoated. Both sets of wells were blocked with 5% (w/v) NFDM in PBST and incubated with various concentrations of NRP1-Fc, followed by the anti-human IgG Fc-HRP antibody. Data are presented as the mean OD 450 ± SD ($n$ = 3 biological replicates). **g** Western blot assay of NRP1, LRP1, pERK and ERK in PSC cells transfected with siControl or NRP1 targeting siRNAs (siNRP1) and treated with PLAU at 100 ng/mL for 72 h. Images are representative of 3 biological replicates. **h** CCK-8 analysis of PSC proliferation modulated by NRP1 knockdown after transfection and PLAU treatment (100 ng/mL). Significance was calculated by the one-way ANOVA with Tukey's post hoc testing and data are presented as the mean ± SD ($n$ = 3 biological replicates). **i** Model for the PLAU-NRP1 mediated PCC-to-PSC intercellular signaling. Source data are provided as a Source Data file.

SEMA3C)[34]. Unlike the PCC-secreted proteins, GO molecular function analysis indicated that the interacting PSC-secreted proteins were highly enriched for extracellular matrix (ECM) structural constituent (Fig. 5g and Supplementary Fig. 15), which is a characteristic feature of active PSCs[35]. The identified ECM proteins, such as collagen I (COL1A1 and COL1A2), fibronectin (FN1), and versican (VCAN), have been reported to bind to CD44[36–38]. Together, our results demonstrate the power of Click-IGC approach for the systematically analysis of paracrine communication between stromal cells and cancer cells in an all-to-all manner.

### Discovery of PLAU-NRP1 as a novel ligand-receptor pair

Having identified PLAU as the most enriched ligand in PCC-to-PSC paracrine signaling, we investigated the functional roles of PCC-secreted PLAU on PSCs. The CCK-8 assay revealed that a low concentration of recombinant PLAU (50 ng/mL) promoted PSC proliferation (Fig. 6a). Western blot analysis showed that recombinant PLAU treatment also led to the activation of AKT and ERK signaling in a dose-dependent manner (Fig. 6b). To confirm whether PLAU is a major factor in PCC-to-PSC paracrine signaling, we silenced PLAU in MIA PaCa-2 and PANC-1 cells using siRNAs. RT-qPCR analysis revealed a

more than 70% decrease in PLAU mRNA expression after siRNA treatment (Supplementary Fig. 16), and western blot analysis also showed a significant decrease in PLAU protein expression (Supplementary Fig. 17). Cell viability of PSC cells either co-cultured with si-PLAU transfected PCCs or cultured in CM of si-PLAU transfected PCCs was significantly reduced compared with sicontrol groups, demonstrating the profound role of PLAU-mediated intercellular signaling from PCCs to PSCs (Fig. 6c). We then performed the Photo-IGC experiment using probe 1-labeled PLAU as the ligand to identify its receptors on PSCs. Excitingly, NRP1 and LRP1, which we had previously discovered by Click-IGC using PCC-CM, were identified as receptors of PLAU (Fig. 6d). In addition, Photo-IGC using N-deglycosylated PLAU also identified NRP1, suggesting that PLAU-NRP1 interaction is independent of the N-glycans of PLAU (Fig. 6e). To determine the PLAU-NRP1 binding affinity, we performed an ELISA binding assay using recombinant PLAU and Fc-tagged NRP1 ectodomain. NRP1 was found to potently bind to PLAU with a dissociation constant (Kd) of 9.5 nM (Fig. 6f). To investigate the role of NRP1 in the PLAU-induced signaling and cell proliferation, we examined the effects of NRP1 knockdown on PLAU-treated and untreated PSCs. Knockdown of NRP1 was confirmed at the mRNA level by RT-PCR (efficacy > 50%) and at protein level by western blot (Fig. 6g and Supplementary Fig. 18a). The activation of ERK by PLAU was abolished by NRP1 knockdown (Fig. 6g). Meanwhile, the expression of LRP1 was not significantly changed after NRP1 was knocked down (Fig. 6g). Furthermore, NRP1 knockdown did not affect cell proliferation in the absence of PLAU after 48 h, but significantly reduced the PLAU-induced cell proliferation (Fig. 6h and Supplementary Fig. 18b). Collectively, these results demonstrate that PCC-secreted PLAU is a novel and functional signaling ligand for NRP1 on the surface of PSCs, and the PLAU-NRP1 interaction mediates the downstream signaling pathways and cell proliferation (Fig. 6g).

## Discussion

Cell-cell communication through ligand-receptor interactions is essential in both normal and pathological processes. With the development of single cell RNA-sequencing (scRNA-seq) technology, several bioinformatics methods have been developed to infer cell–cell interactions from gene expression of ligands and receptors in signal sending and receiving cells using scRNA-seq data[39,40]. The rapidly evolving cell-type-resolved proteomics has also been able to unravel cell–cell interactions at the protein level[41]. Rieckmann et al. deduced the immune cell communication from the quantitative data of total proteome and secretome of flow cytometry-sorted cells by bioinformatic analysis[41]. However, those interaction inference methods largely rely on gene/protein expression level, which cannot identify new interactions and might undervalue signaling complexes containing low-abundant ligands or receptors. Herein, we aim to develop an interaction-centric chemical proteomics strategy to directly uncover the secreted ligands bound to receiving cells and unbiasedly reveal their receptors in an all-to-all manner (Fig. 5).

We demonstrated the versatility of Photo-IGC in identifying both glycosylated (HGF, SARS-CoV-2 RBD) and non-glycosylated (EGF, INS, and PDGF-B) ligands using probe 1, which has a universal ligand conjugation group (NHS ester) and a universal crosslinking group (diazirine). The optimized Photo-IGC workflow results in high specificity and sensitivity, requiring ~1000-fold less ligands and ~10-fold fewer cells than reported ligand-guided receptor capture methods[9–12]. However, probe 1 has a limitation in efficiently labeling low-abundance ligands in complex protein mixture (Figs. 2f, 3f), making it challenging to label low-abundance ligands in CM without over-labeling higher abundance ligands. To address this limitation, we designed probe 2 and probe 3 with an aminooxy group for glycan labeling. Since glycans are less frequently located at protein-protein interaction interfaces than lysine residues[42], the use of excess probe is allowed for labeling both low- and high-abundance ligands without disrupting the ligand-receptor interactions. For example, the low-abundance LIF in KP4-CM can be identified using probe 2 and probe 3, but not probe 1. In addition, compared to probe 2/3, the probe 1 results in higher background binding but less ligand-receptor pairs, suggesting that glycans are more proper labeling positions than lysine residues for crosslinking.

In Photo-IGC, the highly reactive diazirine intermediate provides high selectivity but is prone to quenching by solvent water, leading to reduced crosslinking efficiency. In contrast, the development of probe 3 involved the utilization of bioorthogonal click chemistry, which typically exhibits reaction rates over 10–100 times faster than conventional crosslinking reactions (such as lysine/NHS reaction[43] and the catalyzed hydrazide chemistry[44,45]) at physiological pH. The resulting Click-IGC empowers the identification of low-abundant glycosylated ligand-receptor complexes and the in situ interrogation of cell–cell communication between pancreatic CAFs and cancer cells. Therefore, we recommend the use of probe 3 to study intercellular signaling using CM as ligands. Probe 2-based Photo-IGC could be the alternative when the metabolic labeling with azido sugars or potentially azido amino acids is not applicable (e.g., for tissue samples). For identifying receptors of single ligand, probe 1 is the probe of choice if the purified ligand is available. However, probe 2/3 is useful in the cases that the glycosylated ligand is supplied with high concentration of carrier proteins.

Last but not the least, we demonstrated that IGC methods has a great specificity for exploring receptors of the whole secretome in single IGC analysis. Since current IGC workflow is based on bottom-up proteomics strategy, the direct information of crosslinked ligand-receptor pairs is missing after protein digestion. To this end, we presented the interaction score using the secretome and surfaceome profiles of the analyzed cell pairs. Furthermore, to avoid annotation bias to ligands and receptors with high abundance, we took in consideration of the relative abundance changes of ligands and receptors before and after interaction. Our integrated quantitative proteomic analyses therefore provided an unbiased proteomic technology for profiling interacting receptors and ligands in biological contexts. The applicability of this technology has been validated by the discovery of a novel ligand-receptor interaction in a pancreatic tumor microenvironment model. Collectively, this study paves the way for the investigation of cell–cell communications in tumor microenvironment and other biological systems with limited starting material.

## Methods

### Cell culture and conditioned medium collection
The cancer cell lines HeLa, NIH 3T3, Vero E6, K562, PANC-1 and MIA PaCa-2 were purchased from American Type Culture Collection. KP4 cell line was acquired from JCRB, and human pancreatic stellate cell line PSC was obtained from ScienCell (# 3830). Cells were cultured according to the supplier's instructions. For secretome analysis, cells were grown to ~80% confluence and washed three times with PBS and then cultured in serum-free medium for 24 h. The conditioned medium (CM) was collected and spun down for 5 min at 3,000 g and filtered through a 0.45-μm filter to remove cell debris and contaminating cells. The CM were concentrated 10-fold using a 3-kDa cut-off Amicon Ultra centrifugal filter at 4600 g at 4 °C, and diluted 10-fold with PBS. The concentration step was repeated twice. The protein concentration was determined by using the Pierce 660 nm protein assay (Thermo Scientific).

### Metabolic labeling
To cells at 60% confluency, media supplemented with 10% FBS and 100 μM Ac₄ManNAz, Ac₄GalNAz, Ac₄GlcNAz (Click Chemistry Tools, 1000× stock in DMSO), or DMSO vehicle was added. Cells were metabolically labeled for 24 h and washed with PBS. For Click-IGC experiments, cells were cultured in serum-free media containing 100 μM corresponding sugar analog for an additional 12 h at 37 °C.

## Flow cytometry analysis

Ac$_4$ManNAz labeled and unlabeled K562 cells were collected and washed with PBS. Cells were then resuspended in 320 μL of cold PBS containing 50 μM probe 3 and 200 μg/mL BSA. CuAAC catalyst buffer was prepared by sequentially adding of CuSO$_4$ (Aladdin), THPTA (Click Chemistry Tools), aminoguanidine (Aladdin) and sodium ascorbate (Aladdin) to PBS in a 1:5:20:50 molar ratio, and placed on ice for 10 min before adding to the cells. Then 10 μL of different concentrations of catalyst buffer were added to the cell suspension to make final concentrations of 50 μM, 100 μM and 150 μM of copper ions, and incubated at 4 °C for 15 min. Cells were washed with PBS and incubated with streptavidin-cy3 (1:5000) in staining buffer (PBS containing 0.2% BSA and 0.1% sodium azide, BD Pharmingen) for 30 min at 4 °C. The labeled cells were washed with PBS and analyzed with a FACSAria SORP flow cytometer (BD Biosciences). Data were analyzed using FlowJo (version 10.8.1) software.

## Synthesis of trifunctional probes

Probe 1 was synthesized according to our previous report[17]. Synthetic procedures and characterization of probe 1, probe 2 and probe 3 are detailed in the Supplementary Note 1.

## Preparation of ligand-probe conjugates by NHS ester chemistry

EGF (Sigma–Aldrich, SRP3027; 0.1 μg/μL), HGF (Peprotech, 100-39H; 0.1 μg/μL), PDGF-B (Peprotech, 100-14B; 0.1 μg/μL), SARS-CoV-2 RBD (mFc tag, Sino Biological, 40592-V05H; 0.5 μg/μL), porcine insulin (Aladdin, 12584-58-6; 0.5 μg/μL), or PLAU (His tag, Sino Biological, 10815-H08H-A; 0.2 μg/μL) was mixed with probe 1 (final concentration of 0.16 μg/μL) at a protein-to-probe mass ratio of 1:2 in 50 mM HEPES (pH 8.2) for 10 min at room temperature (RT). The required amount of ligand for three replicates was labeled at once, and aliquoted for each replicate. The ligands required for three replicates were labeled at once and divided into equal amounts for each replicate. Equal amount of BSA (Sangon Biotech), glycine or Tris was used as control. The reaction was quenched by adding glycine or Tris buffer (pH 6.8). The N-deglycosylated PLAU was prepared by incubation of PLAU with 24 units/μL peptide-N-glycosidase F (PNGase F, New England Biolabs, P0704S) in 50 mM HEPES (pH 8.2) for 1 h at 37 °C.

## Preparation of ligand-probe conjugates by oxime ligation

LIF (Symansis, 3014D; 0.1 μg/μL, containing 500-fold HSA as carrier protein in the product) or HGF (0.1 μg/μL, with 500-fold BSA as carrier protein) were oxidized by 2 mM sodium periodate (Sigma–Aldrich) in PBS at 4 °C in the dark for 30 min. The oxidation was quenched by the addition of 4 mM sodium thiosulfate (Sigma–Aldrich). Small molecules in the reaction mixture were removed by centrifugal ultrafiltration using Amicon Ultra filter (3-kDa cut-off). Subsequently, the probe 2/3 and aniline were added to the ligand solution at final concentrations of 200 μM and 50 mM, respectively. The reaction vessel was placed onto a shaker at 37 °C for 1.5 h. Then the unreacted probes and catalysts were removed by centrifugal ultrafiltration.

## Crosslinking on living cells

Cells were washed with cold PBS three times, and incubated with ligand-probe conjugate solution for 10 min at 4 °C. The solution was then removed and replaced with PBS. For photocrosslinking, the cells were UV irradiated in the UVP CL-1000L UV Crosslinker (365 nm) for 5 min at 4 °C. For click chemistry crosslinking, the catalyst buffer was prepared according to previous reports with slight modifications[18,19]. Briefly, CuSO$_4$, THPTA, aminoguanidine and a freshly prepared solution of sodium ascorbate were sequentially mixed and added to PBS to final concentrations of 50 μM, 250 μM, 1 mM, and 2.5 mM, respectively. This catalyst buffer was placed on ice for 10 min and then added to cells for 15 min at 4 °C.

## Cell lysis and pull-down

The cells were washed with PBS three times, and then lysed with a lysis buffer containing 2% (v/v) Triton X-100, 150 mM HEPES pH 8.2, 1.5 mM EDTA, 60 mM 2-chloroacetamide (Sigma–Aldrich), 1 mM phenylmethanesulfonylfluoride (Sigma–Aldrich), 5 μg/mL aprotinin (Amresco), 5 μg/mL pepstatin (Amresco), and 50 U/mL non-restriction nucleases (Beyotime). The lysate was centrifuged at 14,000 g, 4 °C for 6 min, and the resulting supernatant was transferred to a new tube. Streptavidin beads (Cytiva #17511301; 1 μL bead volume for 10$^6$ cells) were added to the supernatant, and samples were placed on an end-over-end rotator for overnight at 4 °C. Beads were then transferred to spin column, washed three times with washing buffer containing 6 M urea, 1% (w/v) SDS, and 50 mM Tris–HCl (pH 7.4), and washed once with 1.5 M NaCl. The beads were then incubated in alkylation buffer (5 mM TCEP, 50 mM 2-chloroacetamide, 0.2 M ammonium bicarbonate, and 0.5 M NaCl) at 37 °C for 60 min and wash three times with 20% ethanol to completely remove detergents. The purified proteins on beads were digested with 8 ng/μL Trypsin (Promega) and 1.6 ng/μL Lys-C (Wako) in 50 mM ammonium bicarbonate at 37 °C overnight on an end-over-end rotator. The digestion mixtures were acidified, desalted with StageTips, and lyophilized.

## LC−MS/MS analysis

Peptide samples were dissolved in a solution of 5% acetonitrile (ACN) and 4% formic acid (FA). For IGC experiments using purified ligands, samples were analyzed by an EASY-nLC 1000 (Thermo Scientific) chromatography system coupled to a Q-Exactive mass spectrometer (Thermo Scientific). Peptides were separated by an in-house packed column (100 μm i.d. × 20 cm, ReproSil-Pur C18-AQ, 1.9 μm, 120 Å, Dr. Maisch GmbH) with a binary buffer system of 0.1% FA in water (buffer A) and 0.1% FA in ACN (buffer B) at a flow rate of 300 nL/min with an effective gradient from 6% to 22% of solvent B over 42 min, followed by 22% to 35% of buffer B over 8 min. Separated peptides were analyzed with one full scan (350–1500 m/z, = 70,000 at 200 m/z) and 3 × 10$^6$ automatic gain control (AGC) target. Upto 10 most intense ions (1 < z < 6) were sequentially selected with an isolation width of 2.0 Th, 30 s dynamic exclusion and fragmented by higher-energy collisional dissociation (HCD) at 17500 resolution and a normalized collision energy (NCE) of 27. For IGC experiments using secretome as ligands, more sensitive mass spectrometers were used (see Supplementary Note 2).

## Data analysis

Raw files were searched using MaxQuant (version 1.6.14)[46]. UniProtKB human proteome database (UP000005640; released in January 2020), UniProtKB mouse proteome database (UP000000589; released in December 2020), and UniProtKB *Chlorocebus sabaeus* proteome database (UP000029965; released in March 2020; appended with sequences of mFc tagged SARS-CoV-2 RBD) were used for database search of samples from human cells, NIH 3T3 cells and Vero E6 cells, respectively. Contaminants were included in the search, and the reverse database was used to determine the false discovery rate (FDR). Both peptides and proteins were filtered at 1% FDR. Cysteine carbamidomethylation was set as a fixed modification. Methionine oxidation, protein N-terminal acetylation and asparagine/glutamine deamidation were set as variable modifications. The LFQ (minimum ratio count 1, normalization type none) and intensity-based absolute quantification (iBAQ) were enabled to evaluate protein abundances[47]. The "match between runs" feature was activated with a match window of 0.2 min.

Data were processed using R software (version 4.0.2)[48]. Proteins marked as reverse hits, potential contaminants, "only identified by site", and containing fewer than two razor and unique peptides were filtered out. The reviewed and first protein entry in each protein group was selected as representative. Proteins identified "by MS/MS" in all

three replicates in at least one experimental group were selected for the quantification. For proteins only identified "by matching" between groups and had 5-fold smaller intensities than the other group, the LFQ intensities were replaced with the summed peptide intensities to stabilize large LFQ ratios. The LFQ intensities were log$_2$-transformed. For streptavidin pull-down samples, LFQ intensities were normalized to the summed-up LFQ intensities of the endogenous biotinylated carboxylases (ACACA, PC, MCCC1, PCCA and ACACB) and then median normalized. Missing values were imputed from normal distribution with a width of 0.3 and a down shift of 1.8. Statistical analysis was performed with the limma R package (version 3.44.3)[49]. An empirical Bayes moderated $t$-test was used for two group comparisons and $p$-values were FDR adjusted with the Benjamini-Hochberg method. Proteins with an adjusted $p$-value < 0.05 and a |log$_2$ fold change | > 1 were considered significant.

Each protein's mean iBAQ value was normalized by subtracting the mean iBAQ of control samples and then used to calculate the relative iBAQ (riBAQ) value[50,51]. The presence of ligand-receptor interaction should lead to changes in the relative abundance of secreted protein after incubation with cells or that of surface proteins after crosslinking. Therefore, we generated the interaction score as defined in Eq. (1) to assess the possibility of a cell surface/secreted protein acting as an interactor.

$$\text{Interaction score} = \log_2\left(\frac{\text{riBAQ}}{\text{riBAQ}_{ref}} + 1\right) \qquad (1)$$

Cell surface/secreted proteins with significant differences in the volcano plot analysis were used to calculate the interaction score. The riBAQ values in surfaceome data or secretome data were used as references (riBAQ$_{ref}$) for the secreted proteins and surface proteins, respectively.

Gene Ontology enrichment analysis was carried out using DAVID 2021[52].

## Western blotting
Cells were lysed in 2× Laemmli loading buffer and denatured at 95 °C for 5 min. Then proteins were separated on 10% SDS-PAGE gel and transferred onto PVDF membranes. The membranes were blocked in 5% BSA or non-fat dried milk (NFDM) in TBS with 0.1% Tween20 (TBST) at RT for 1 h, followed by incubation at 4 °C overnight with primary antibodies: STAT3 (CST, 9139 s, 1:1000), phospho-STAT3 (CST, 9145 s, 1:1000), ERK (CST, #4695, 1:2000), phospho-ERK-Thr202/Tyr204 (CST, #9101, 1:2000), AKT (CST, 4685 s, 1:2000), phospho-AKT-S473 (CST, 4060 s, 1:2000), NRP1 (Abcam, ab81321, 1:1000), PLAU (Abcam, ab24121, 1:1000), LRP1 (Abcam, ab92544, 1:1000), and β-actin (Beyotime, AF0003, 1:5000). After washed with TBST for three times, membranes were incubated with HRP-conjugated goat anti-mouse IgG (Beyotime, A0216, 1:1000) or goat anti-rabbit IgG (Beyotime, A0208, 1:1000) at RT for 1 h. After washing three times with TBST, blots were detected with Clarity Western ECL Substrate (Bio-Rad) through a Gel Imaging system (Tanon 6100 C) or an Odyssey infrared scanner (LICOR Bioscience).

## Cell proliferation assay
Cell viability was assessed using the CCK-8 assay (MCE) according to the manufacturer's instructions. PSC cells were seeded in a 24-well plate, and the optical density (OD) values (450 nm) were measured after treatment with PLAU or PBS for 0, 24, 48, and 72 h.

## RNA extraction and quantitative real time PCR (RT-qPCR)
Total RNA was extracted from cultured cells with the Eastep Super Total RNA Extraction Kit (Promega) according to the manufacturer's instructions. The first-strand cDNA was synthesized with oligo(dT) and random primers using the High-Capacity cDNA Reverse Transcription Kit (Thermo Fisher Scientific). The mRNA levels of indicated genes were quantified by real time qRT-PCR using the TB Green Premix Ex Taq II reagent (Takara) on the CFX96 system (Bio-Rad). Data were analyzed using GraphPad Prism (version 5.0). The $2^{-\Delta\Delta Ct}$ method was used to quantify the relative RNA expression level, and β-actin served as an endogenous reference. All primers used are listed in Supplementary Table 1.

## siRNAs transfection
Cells were cultured in 6-well plates for transfection. For each well, 7.5 μL of RNAiMAX (Life Technologies) diluted in 250 μL of Opti-MEM (Sigma) and 25 pmol of siRNA (RiboBio) diluted in 250 μL of Opti-MEM were mixed and incubated for 15 minutes, and then added to the cells. All siRNAs used are listed in Supplementary Table 2.

## Co-culture system
PANC-1 or Mia PaCa-2 cells with siControl or siPLAU were placed in the upper chamber with a 0.4-μm pore size (Corning, #3412). PSC cells were seeded in a 24-well plate, and cells were co-cultured for 48 h. Then, PSC cell proliferation was detected using the CCK-8 assay. In another experiment, PANC-1 or Mia PaCa-2 cells with siControl or siPLAU were cultured in 6-well plates for 48 h for knockdown, and then incubated in serum-free medium for 24 h. Then, CM of PCCs was collected for further experiments. PSC cells treated with siControl or siPLAU pre-treated PCCs CM in a 24-well plate, and the OD values were measured at 48 h.

## Solid-phase binding assay
High-binding ELISA plates (Corning, #3690) were coated with s-tagged PLAU (Supplementary Note 3) at a concentration of 1.5 ng/μL (25 μL per well) in PBS overnight at 4 °C. The uncoated wells were used as controls. Both sets of wells were blocked with 100 μL of 5% (w/v) NFDM in PBST (PBS with 0.05% Tween-20) for 2 h at RT. After removal of the NFDM solution, serial dilutions of NRP1-Fc (Sino Biological, 10011-H02H) in 5% BSA were added and incubated for 2 h at RT. After washing with PBST, wells were incubated with HRP coupled goat anti-human IgG Fc antibody (Beyotime, 1:1000 dilution in 5% NFDM) solution for 1 h at RT. Wells were washed with TBST, and 25 μL per well of TMB One-Step Substrate Reagent (RayBio) was added to detect binding, followed by the addition of stop reagent (0.2 M H$_2$SO$_4$). Absorbance was measured at 450 nm using a BioTek microplate reader. ELISA measurements were carried out in triplicate.

## Data availability
The mass spectrometric raw data are deposited on ProteomeXchange via the PRIDE partner repository[53] with the dataset identifier PXD038018. All other data that support the findings of this study are provided in the Supplementary Information/Source Data files. Source data are provided with this paper.

## Code availability
Custom processing code used for MS data analysis and figure generation is available as Supplementary software.

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

## Acknowledgements

This work is supported by grants from the China State Key Basic Research Program Grants (2021YFA1302603, 2022YFC3401104, 2020YFE0202200, 2021YFA1301601, and 2021YFA1301602), the National Natural Science Foundation of China (32201218, 22125403, 91953118, 32171433, and 22104047), the Shenzhen Innovation of Science and Technology Commission (JCYJ20200109141212325, JCYJ20210324120210029 and JCYJ20200109140814408), and Guangdong province (2019B151502050).

## Author contributions

J.Z. and R.T. designed the project. J.Z. and Z.Z. performed proteomics experiments. J.Z. conducted biochemistry experiments and bioinformatics analyses. Z.Z. performed chemical probe synthesis. C.F. designed and performed cell biology and molecular biology experiments. Y.W. and A.H. provided proteomics data analysis support. X.Y. performed the flow cytometry analysis. W.G. reviewed the figures. R.T. conceived the original idea and supervised the research. J.Z. and R.T. wrote the manuscript with input from all authors.

## Competing interests

The authors declare no competing interests.
