## [Peer Review File · Nature Communications]

Reviewers' Comments:

Reviewer #1:

Remarks to the Author:

General comments:

In this manuscript, "Deciphering intercellular signaling complexes by interaction-guided chemical proteomics", the authors describe three trifunctional crosslinkers, one of which (Probe 1) has been previously published. The manuscript shows several applications of these crosslinkers, including an interesting all-to-all comparison of a complex mixture of secreted proteins from conditioned media (serum free only) to a cell line. The crosslinker design and application are interesting, but in my opinion, the examples and discoveries shown here fall short the breadth of applications and biological insight found in other Nature Communications publications in this field (i.e. Reference #10, #15 in this manuscript). There are also significant concerns about the experimental design and a lack of information provided for each experiment.

Major concerns:

1. The authors claim that the IGC approach significantly outperformed any related existing method is not well supported. Nearly all of the experiments shown in this manuscript used 1 million or more cells, an amount that was also shown to work well with HATRIC (Reference 10 in this manuscript). While the ability to use less ligand could be useful, I would consider this an incremental improvement, especially since ligand-labelling with crosslinkers cannot easily be performed at very small scale (50 ng) anyway (at least without using special microcapillary reaction systems). The volumes are simply too small to work with and conjugation efficiency goes down with lower concentration. In fact, the authors of this manuscript fail to state how much ligand was used in their ligand-labelling reaction.

2. In the introduction, the authors' state (line 56-59) that a large number of previous crosslinking or proximity labelling methods are "hypothesis-driven and best suited for studying the interactomes of the protein of interest". However, most of these studies are not "hypothesis-driven" and many are looking at equally complex interactions as is shown here.

3. The authors statement that they "verified that PLAU is a novel and functional signaling ligand for NRP1" is not strongly supported. The authors did show that PLAU can bind to NRP1 ectodomain on PSC cells, and that PLAU plays a role in PSC cell growth and viability. However, they did not adequately demonstrate that these functional effects are mediated by PLAU binding to NRP1 and not by PLAU binding to one of the more classical PLAU receptors such as LRP1 (or any of the other receptors that came up as "hits" in Fig 6d or 6e).

4. Inadequate experimental details are provided for nearly all experiments. A few examples, though there are too many to list:

- a. How many repeats were performed for each panel in Figure 2?
- b. In the preparation of the ligand-probe conjugates, how much ligand was labelled? At what concentration was the ligand? Why was BSA used sometimes but not others?
- c. How much recombinant SARS-CoV-2-RBD-mFc was used in Fig 2e? And for every other experiment.
- d. How many cells and how much HGF and/or BSA was used for Fig 2f? etc.
- e. All bar charts showing MS/MS counts are unclear. Was this MS/MS counts in the crosslinked sample only? How many counts were found in the negative control arm of that experiment?
- f. What was the protein concentration in the conditioned media when performing conjugation with probes? Was it left in concentrated form, or was it diluted back to be similar to the conditioned media?
- g. What concentration was the ligand-probe conjugate solution when incubated with the cells? Was the ligand always diluted to 1 mL no matter how many cells or how much ligand was being used?

5. Why are different negative controls used for nearly every IGC experiment? What effect does the control arm have on the results? Sometimes, glycine is used (Fig 2b). Other times, Tris or BSA is used. In Figure 4d/5c, catalyst +/catalyst - is reported. In Fig 5b, cCM+/cCM- is used, in Supp Fig 10, UV+ vs. UV- is used. What guides this selection of the negative/comparison control? Does using BSA as a diluent make the results look "nicer" at lower levels than it would with an unrelated negative control (e.g. glycine)? While the different probes may influence some of these choices, the apparently random selection of controls with no discussion around how or why these comparisons were selected, significantly detracts from the usefulness of these approaches for other researchers.

6. The data is often presented in a different way for each experiment. For example, in Supp Fig 10 – why is the volcano plot one-sided? Were no proteins found to be higher in the UV- than in the UV+ samples? With such a diverse way of designing and reporting experiments, the potential for these crosslinker designs to be useful for other researchers in other applications is greatly diminished.

7. It is somewhat surprising that so few proteins were identified in these experiments (based on the number of dots in the volcano plots). Do you see background binding with Probe 1? How does this compare to Probe 2/3? For each volcano plot, an excel table containing protein information (# peptides identified, quantification in each sample, summary stats, etc) should be provided as supplementary information. As it is, only the few points that the authors chose to label are “identified” for the reader. For example, what are the gray, significantly changing dots in Fig 4d or Supp Fig 10? Why were they identified in the experiment?

8. The interaction score was poorly explained. By subtracting the “control” iBAQ value (what samples are the control, anyway??), does this bias for highly expressed proteins? Do the different mass spectrometers used (with different dynamic ranges etc) for the cell surface profiling and the IGC experiments affect the results? Do different MS instruments favor different peptides?

9. More discussion (and perhaps data) to compare the three different probes is needed to provide the reader with a comparison of how they work and an indication of how to best choose the probe for each application. When labelling conditioned media proteins/protein mixtures, how efficient is the labelling with Probe 1 and Probe 2? Do they look similar? How do you explain differences between panel a and b in Supp Fig 10? How different are these two probes that should only differ in how they are ligated to their ligands?

10. How many PSC cells were required for all of the experiments described in Figure 5, including surface mapping and PSC conditioned media collection? What is the minimum number of cells required for all-to-all mapping?

11. In general the discussion section could be greatly strengthened to comment on the strengths and weaknesses, background binding, etc seen with each of these probes and workflows (what is the role of BSA as a “carrier protein”, how can the interaction score and be used to find new paracrine systems, what are the benefits and drawbacks of each probe? etc.)

12. The “double” normalization, first to native biotinylated proteins, then to median, seems questionable. If there are many less proteins in the control, it seems that this second normalization could be quite harsh. It is also unclear when missing values are being imputed. Earlier, it states that all proteins were identified in all technical replicates before quantification. Why is there then the need to impute values? Did you impute values at the peptide or protein level? Explain why imputation is valid here.

13. What was the control in the NRP1 ELISA?

Minor comments:

1. Several minor grammatical errors throughout the manuscript that could likely be corrected during proofing stage
2. Full western blots (with full molecular weight range) should be shown uncropped in the supplementary information.
3. Does copper affect cell viability?

Reviewer #2:

Remarks to the Author:

Zheng and colleagues in their article with the title “Deciphering intercellular signalling complexes by interaction-guided chemical proteomics” describe an efficient method how to identify novel ligand-receptor pairs. They utilize three different trifunctional linkers offering certain degree of flexibility how to cross-link the ligand-receptor pair based on their close proximity interaction. Although various

chemical proteomic protocols for identification of protein-protein interactions have been developed, the application for analysis of low abundant ligand-receptor interactions is relatively novel. The presented efficiency of the enrichment is impressive and superior to previous publications. However, although the experiments are described and documented well it is not clear what is the main principle(s) which allowed authors to improve the method. The practical efficiency to find new ligand-receptor pairs is relatively limited to one new interaction.

Major corrections:

The authors must provide a quality control of the MS data. At least, the histograms showing the number of imputed values and Pearson correlations of protein intensities between the replicates. For the main "hits" the supporting information should contain the profile plots, before imputation representing the actual LFQ intensities.

The usage of as low as 1 μL of the streptavidin beads suspension for the enrichment of 10^6 cells is surprisingly low. Majority of the protocols would use $>20 \mu\text{L}$. Is this a crucial aspect of the protocol's success? How was the beads amount optimized? It might be very challenging to work with such low beads amount, but it might be the critical point for the protocol's sensitivity.

The identification of the novel PLAU receptor NRP1 is certainly of high importance. However, it is not possible to conclude from the provided data, if the NRP1 is actually the major receptor. The direct biochemical comparison of the two receptors is not provided. The enrichment of the ligand-receptor itself (visible in the volcano plot) is always relative depending on the labelling efficiency with the probe, which might be different for the two receptors, degree of the background binding or imputed values. Here, the profile plots before imputation might help to review. The relative higher enrichment of the NRP1 not necessarily reflects the actual ratio on the cells.

Although, the acceleration of PSCs proliferation by PLAU is an interesting finding, it is not shown if it is actually linked with the newly discovered NRP1.

It would be beneficial for the field and those applying the presented protocol to have better information how the probes efficiency compares. Is it possible to enrich the PLAU-NRP1 with Probe 1 and 2?

Fig 4h: The loading control should be added.

Fig 2a: Why the authors did not use as the control not UV irradiated EGF-probe conjugate?

Line 222: In conclusion, it should be clearly stated what are the principal mechanisms behind the method sensitivity and fidelity. The authors should provide the reader with comparison of the probes 1 – 3 to help navigate the selection of the suitable probe for further applications.

Line 495: It is not possible to access the original data on PRIDE repository. This should be corrected.

Minor corrections:

Line 102: correct the typos

Line 208: Please quantify "almost complete silencing"

Line 362: What is the final concentration of at least one component in click reaction mixture?

Line 373: Similarly, please add the exact final concentration of the protein or probe.

Line 396: What type of the beads were used? Streptavidin-coated magnetic or agarose beads, it would be beneficial to include the product number from the Thermo catalogue.

Line 429: two razor and unique peptides were needed to confidently assign the protein. Does it mean that two razor peptides are sufficient as well? Standardly, at least 1 unique peptide should be required

for confident identification of the protein. Still, 1 unique peptide might lead to biased quantification.

Reviewer #3:

Remarks to the Author:

In this manuscript, the authors develop a trifunctional probe that takes advantage of glycan labeling and click chemistry to probe interactions between secreted ligands and their receptors. They deploy these probes for analysis of single ligands and for all-vs-all analysis of the secretome from one cell type and cell surface receptors from another cell type after crosslinking. The authors identify a new receptor-ligand interaction between NRP1 and PLAU. Although the IGC approach can probe receptor-ligand interactions, it does not seem to directly connect them in a 1-to-1 fashion in the absence of prior knowledge or candidate-based approaches. I was unable to determine, for example, how the authors chose to follow up on the PLAU-NRP1 interaction as it seems there were many other possible pairs in the dataset. Although the authors claim that HATRIC requires more material, the paper that they cite also includes experiments that use only one million cells, and the cell types used differ, so the experiments are not directly comparable. Both HATRIC and IGC ligands can be synthesized, so their accessibility seems to be the same despite the commercialization of HATRIC. IGC is therefore a potentially complementary technology to existing methods but is unlikely to displace them. The manuscript was a bit difficult to follow and would benefit from substantial revision.

Reviewer #4:

Remarks to the Author:

The work by Zheng and coworkers, represents a smart approach to cover interactions of even lower abundant proteins. Although none of the used functionalities or techniques is new per se, its combination including the synthesis of trifunctional crosslinkers is innovative and allows to profile inter-cell interactions of surface receptors within their native environment – meaning across living cells. The need of led ligands and less cell material is the biggest advantage over other receptor capture techniques and over classical crosslinking mass spectrometry approaches.

On the downside it seems that the workflow shows its full performance only in case of glycosylated proteins, which limits its applicability.

Overall, the manuscript is well written and will be of value for the community once published. Please find some comments below:

- The authors mention that their spacer arm length of 60 Å is well suited for inter-cell crosslinking. I understand that longer spacer arms will capture more interactors thanks to their extended range and flexibility. However, this might bear the risk to capture of non-specific random interactors that are within the extended linker range within solution and common linker reagents for classical crosslinking mass spectrometry, like, DSS, DSSO, DSBU, etc., are in the range of 10 -20 Å long. Can you comment on these thoughts and especially how you'd ensure to capture only specific interactors?
- If I understood correctly, in Figure 3C, the authors show by flow cytometry the efficiency of the click chemistry. They nicely show that in the absence of the catalyst Cu(I) no binding takes place as this looks like their control. I was wondering about two things: 1st, what is the control? I guess these are unlabeled cells, but how much Cu(I) was used there? Sorry if I just missed this info. 2nd, it's a bit contra-intuitive at first sight that 50 µM Cu(I) showed a better performance than 100 or 150µM. Can you comment on potential reasons? Have you repeated this experiment to validate its outcome? Since the catalyst is not used up during the reaction, I would have assumed the difference between Cu(I) concentrations should be more of kinetic nature, with improved reaction speed the higher the concentration.

General comment:

IGC seems very well suited to detect interaction partners, with higher sensitivity compared to standard affinity purification or other methods as described by the authors. However, since a crosslinker is used, IGC could be even way more powerful in case the actual interaction site could be identified. (At least for future studies) It should be possible to elute the biotin-crosslinked peptides off the streptavidin beads after digestion (e.g. by competition with free biotin, or by introduction of a cleavage group into the linker) and yield a purified sample of the crosslinked peptides for MS analysis. With this the degree of information obtained from IGC experiments would be clearly improved.

However, I see that such additional experiments might be more effort than possible in the given time for a revision, especially in case a new adopted probe with an additional cleavage site must be synthesized.

Minor remarks:

- There seem quite some typos or grammar errors distributed over the manuscript. I suggest another proofreading, maybe from a native English speaker. Some that I found:
 - o Line 44: "...that are not necessarily happen in biological microenvironment." Seems grammatically incorrect to me
 - o Line 66: was/were seems mixed up to me
 - o Line 102: Seems also grammatically incorrect/not a proper sentence "...since protein is nearly always contains..."
- In Supplemental Figure 18, the figure caption does not fit to the labeling. It seems A) and B) is mixed up.

General Responses:

First of all, we appreciate all the reviewers for their diligent examination of the manuscript and critical comments and suggestions. To streamline our response and prevent redundancy, we will address several frequently raised comments from the reviewers in this section.

1. The advantage of the IGC approach over existing methods:

To the best of our knowledge, the IGC approach is the first feasible approach to discover ligand-receptor complexes in cell co-culture system. There are no proteomic methods available to discover signaling ligands in cell-conditioned media or to identify receptors on living cells without prior definition of ligands. As for identifying the receptors for the pre-defined ligand, the pioneered and best available method is HATRIC (*Nat. Commun.* 2018, 9, 1519) which requires 100 µg of ligand, typically over 3 orders of magnitude above the endogenous level. In contrast, various examples in our study (Fig. 2 and Fig. 3) demonstrate that the IGC method requires ~1000-fold less ligands. Moreover, as suggested in the original paper and its commercial service instructions, the HATRIC typically requires 20 million cells per replicate, which is too expensive for studying physiologically relevant systems (e.g., primary cells). In contrast, we routinely applied the IGC approach to identify receptors on 1 million cells, including the real application involving primary stromal cells (Fig. 5). Compared to a HATRIC case using 1 million cells, the IGC approach achieved receptor identification on 0.1 million cells. As summarized in the abstract, “the unparalleled sensitivity and selectivity allow systematic crosslinking and identification of ligand-receptor complexes formed between cell secretome and surfaceome in an unbiased and all-to-all manner”.

2. The comparison of three probes and the mechanism behind the method sensitivity and fidelity:

We addressed this concern by adding the following statement to the Discussion section:

However, probe 1 has a limitation in efficiently labeling low-abundance ligands in complex CM without over-labeling higher abundance ligands (Fig. 2f, 3f). To address this limitation, we designed probe 2 with an aminooxy group for glycan labeling. Since glycans are less frequently located at protein-protein interaction interfaces than lysine residues⁴², the use of excess probe is allowed for labeling both low- and high-abundance ligands without disrupting the ligand-receptor interactions. For example, the low-abundance LIF in KP4-CM can be identified using probe 2, but not probe 1. In addition, compared to probe 1, the probe 2 leads to lower background binding and identifies more ligand-receptor pairs, suggesting that glycans are more suitable labeling positions than lysine residues for crosslinking (Fig. 3e, 3f). On the other hand, although the highly reactive and short-lived intermediate of diazirine enables high selectivity of Photo-IGC, its quenching with solvent water also decreases the crosslinking efficiency. To further increase sensitivity, Probe 3 was developed to crosslink via the highly efficient CuAAC click chemistry, which is usually 10–100 times faster than hydrazide chemistry (e.g., HATRIC) at physiological pH^{43, 44}. The resulting Click-IGC empowers the identification of low-abundant glycosylated ligand-receptor complexes and the in-situ interrogation of cell-cell communication between pancreatic CAFs and cancer cells. Therefore, we recommend the use of probe 3 to study intercellular signaling using CM as ligands. Probe 2-based Photo-IGC could be the alternative when the metabolic labeling with azido sugars or potentially azido amino acids is not applicable (e.g., for tissue samples). For identifying receptors of single ligand, probe 1 is the probe of choice

if the purified ligand is available. However, probe 2/3 is useful in the cases that the glycosylated ligand is supplied with high concentration of carrier proteins. Last but not the least, the use of a very small amount of streptavidin beads for affinity enrichment from limited sample also significantly increase the sensitivity (Fig. R6).

3. Lack of strong support that PLAU's functional effects are mediated by NRP1 binding:

To investigate the role of NRP1 in the PLAU-induced signaling and cell proliferation, we examined the effects of NRP1 knockdown on PLAU-treated and untreated PSCs. Knockdown of NRP1 was confirmed at the mRNA level by RT-PCR (efficacy > 50%) and at protein level by western blot (Fig. 6g and Supplementary Fig. 18a). The activation of ERK by PLAU was abolished by NRP1 knockdown (Fig. 6g). Furthermore, NRP1 knockdown did not affect cell proliferation in the absence of PLAU after 48 h, but significantly reduced the PLAU induced cell proliferation (Fig. 6h and Supplementary Fig. 18b). These additional experiments provided strong evidence to support our claim that PLAU is a novel and functional signaling ligand for NRP1, which has been included in the revised manuscript. Last but not the least, NRP1 was found to have over 10-fold higher LFQ intensity than LRP1, suggesting NRP1 could play a more important role than LRP1 in PLAU-induced signaling (Supplementary Fig. 46).

New data added into the revised manuscript:

Here is a brief summary of the experiments we carried out in response to the reviewers' comments to further test our original conclusions. Please refer to the point-by-point responses for detailed description.

- NRP1 knockdown experiments (Fig. 6g and Supplementary Fig. 18a).
- Western blot assay of the activation of ERK by PLAU in PSCs with or without NRP1 knockdown (Fig. 6g).
- Cell proliferation assay of NRP1 knockdown PSCs in the absence of PLAU after 48 h (Fig. 6h and Supplementary Fig. 18b).
- Photo-IGC experiments for the identification of receptors of HGF using BSA, glycine, and Tris as negative controls, respectively (Supplementary Fig. 8a-c).
- Photo-IGC experiment for the identification of receptors of EGF using BSA as the negative control (Fig. R3a).
- Photo-IGC experiment for the identification of receptors of PDGFB using BSA as the negative control (Fig. R3b).
- AP-MS analysis of the biotinylated proteins from the lysates of 1 million HeLa cells with 1 μ L, 5 μ L, and 10 μ L of streptavidin beads, respectively (Supplementary Fig. 8g, h).
- Photo-IGC experiment for the identification of receptors of EGF through UV+/UV- comparison (Fig. R7).
- The repeated flow cytometric analysis of Ac4ManNAz labeled and unlabeled K562 cells incubated with probe 3 in the absence or presence of different concentration of copper catalyst (Fig. 3c).

Point-by-Point Responses to individual comments:

Response to the comments raised by Reviewer 1:

In this manuscript, “Deciphering intercellular signaling complexes by interaction-guided chemical proteomics”, the authors describe three trifunctional crosslinkers, one of which (Probe 1) has been previously published. The manuscript shows several applications of these crosslinkers, including an interesting all-to-all comparison of a complex mixture of secreted proteins from conditioned media (serum free only) to a cell line. The crosslinker design and application are interesting, but in my opinion, the examples and discoveries shown here fall short the breadth of applications and biological insight found in other Nature Communications publications in this field (i.e. Reference #10, #15 in this manuscript). There are also significant concerns about the experimental design and a lack of information provided for each experiment.

Response:

We appreciate for the positive comments from the reviewer and pointing out those two Nature Communications publications (Ref #10 and #15) which are important works in this field. However, those methods are not applied to identify ligand-receptor interactions between secreted ligands and their plasma membrane receptors in all-to-all manner, which is the major goal of our manuscript. In Ref #10, the HATRIC method was successfully applied to identify receptors for various ligands, while it requires 100 µg of ligand and 20 million cells per replicate. It is impractical or expensive to obtain such large amount of secreted ligands in conditioned media and primary cells. In Ref #15, LUX-MS is a type of proximity labeling method, which in principle and based on our own experience (*Nature Communications*, 2021, 12, 71) has a relatively large labeling radius. Although the sensitivity is improved (still needed 10~20 million cells per replicate), many non-specific binding proteins were also identified even using one pure ligand (Ref #10). When using CM (the whole set of secreted proteins) as ligands, there will be much more false positives that makes it difficult to find real receptors. To this end, our work aimed to significantly increase the labeling selectivity and analysis sensitivity by developing new trifunctional probes and chemical proteomic workflow. More importantly, we applied them to investigate ligand-receptor interactions in an all-to-all manner and pancreatic cancer systems.

To address the reviewer’s comments, more experimental details and biological validation are provided in the revised manuscript. And more discussion on the experimental design has been added.

1. The authors claim that the IGC approach significantly outperformed any related existing method is not well supported. Nearly all of the experiments shown in this manuscript used 1 million or more cells, an amount that was also shown to work well with HATRIC (Reference 10 in this manuscript). While the ability to use less ligand could be useful, I would consider this an incremental improvement, especially since ligand-labelling with crosslinkers cannot easily be performed at very small scale (50 ng) anyway (at least without using special microcapillary reaction systems). The volumes are simply too small to work with and conjugation efficiency goes down with lower concentration. In fact, the authors of this manuscript fail to state how much ligand was used in their ligand-labelling reaction.

In fact, in the HATRIC paper, the authors demonstrated they can identify EGFR and TFR1 as the receptors of anti-EGFR antibody and Holo-transferrin (TRFE) from 1 million MDA-MB-231 cells per replicate. However, all other experiments were conducted using 20 million cells. In contrast, we can identify HGF receptor (MET) from 0.1 million HeLa cells with Click-IGC method, and all other Click-IGC experiments required 1 million cells. Therefore, the IGC method is better than others in terms of the required amount of cells.

The ability to use less ligand is a critical requirement for studying paracrine communication using conditioned media as real samples, since the availability of secreted ligands in conditioned media is often limited. To our understanding, manually performing the small-scale ligand labeling experiment is not difficult. The resulting ligand-probe conjugates can be diluted and aliquoted for each replicate of the IGC experiment. All liquid handling can be conducted with commonly used Eppendorf pipettes (pipette volume size: 0.1-2.5 μ L, 2-20 μ L, and 20-200 μ L). As suggested, the amount of ligands used were added in the figure legends, and more details of ligand labeling is added in the Methods section of the revised manuscript. For example, in the IGC experiments with 50 ng of EGF shown in Fig. 2b, 1.5 μ L of 0.1 μ g/ μ L EGF (150 ng for three replicates) was mixed with probe 1 at a protein-to-probe mass ratio of 1:2 in 50 mM HEPES (pH 8.2) for 10 min at room temperature (RT). Then the ligand-probe conjugate solution was divided into equal amounts for each replicate.

2. In the introduction, the authors' state (line 56-59) that a large number of previous crosslinking or proximity labelling methods are "hypothesis-driven and best suited for studying the interactomes of the protein of interest". However, most of these studies are not "hypothesis-driven" and many are looking at equally complex interactions as is shown here.

We are sorry for the misunderstanding. In line 56-59, we stated that the proximity labelling methods, such as PUP-IT, μ Map, LUX-MS, and PhoTag, are hypothesis-driven and best suited for studying the interactomes of the protein of interest. In these studies, the researcher decided in advance of the experiment what proteins to study. Then the novel enzyme tags or catalyst-antibody conjugates were fused/bound to the protein of interest (POI) to study the interactomes of POI. As we well presented in Fig. 4-5, we aimed to study intercellular signaling without prior knowledge of what proteins are important or functional in the paracrine signaling. Although those previous studies could reveal complex interactions, they are "hypothesis-driven" in the sense that they require the pre-determination of the POIs.

3. The authors statement that they "verified that PLAU is a novel and functional signaling ligand for NRP1" is not strongly supported. The authors did show that PLAU can bind to NRP1 ectodomain on PSC cells, and that PLAU plays a role in PSC cell growth and viability. However, they did not adequately demonstrate that these functional effects are mediated by PLAU binding to NRP1 and not by PLAU binding to one of the more classical PLAU receptors such as LRP1 (or any of the other receptors that came up as "hits" in Fig 6d or 6e).

We acknowledged reviewer's comment. As explained in detail in General Response #3, we performed additional experiments and examined the effects of NRP1 knockdown on PLAU-induced PSC cell signaling and cell proliferation. The efficacy of NRP1 knockdown was confirmed (Fig.

R1a, b). NRP1 knockdown abolished the activation of ERK (Fig. R1b) and significantly reduced the PLAU induced cell proliferation (Fig. R1c). These additional experiments provided strong evidence to support our claim that PLAU is a novel and functional signaling ligand for NRP1, which has been included in the revised manuscript.

Fig. R1 a, PSCs transfected with negative control siRNA (siControl) or NRP1 siRNA (siNRP1) were subjected to RT-qPCR. The *P* values were calculated using two-sided Student's *t*-test and data are presented as the mean \pm SD ($n = 3$); **** $P < 0.0001$. **b**, Western blot assay of NRP1, pERK and ERK in PSC cells transfected with siControl or NRP1 targeting siRNAs (siNRP1) and treated with PLAU at 100 ng/mL. **c**, CCK-8 analysis of PSC proliferation modulated by NRP1 knockdown after transfection and 48 h of PLAU treatment (100 ng/mL). Significance was calculated by the one-way ANOVA with Tukey's post hoc testing and data are presented as the mean \pm SD ($n = 3$). ** $P < 0.01$, *** $P < 0.001$, **** $P < 0.0001$; NS, not significant.

4. Inadequate experimental details are provided for nearly all experiments. A few examples, though there are too many to list:

We appreciate for this comment and systematically revised our manuscript accordingly to provide more detailed descriptions of our experiments.

a. How many repeats were performed for each panel in Figure 2?

In fact, all experiments in Figure 2 were performed in triplicates per condition. To make it clearer, we moved the statement to the end of the figure legend. Accordingly, we have carefully checked all our experimental details and improved our description.

b. In the preparation of the ligand-probe conjugates, how much ligand was labelled? At what concentration was the ligand? Why was BSA used sometimes but not others?

The amount of ligand used for each replicate was added in the revised figure legends, and a slightly more than three times amount of ligand (for three replicates) was used in the preparation of the ligand-probe conjugates. Typically, the concentration of ligand (e.g., EGF, HGF, insulin, and PDGFB) was 0.1 $\mu\text{g}/\mu\text{L}$, and 2~20 μL of ligand were used for labeling. The resulting ligand-probe conjugates were diluted and aliquoted for each replicate of IGC experiments. The reason of using

BSA is explained in the following response (response to question 5).

c. How much recombinant SARS-CoV-2-RBD-mFc was used in Fig 2e? And for every other experiment.

In Fig 2e, 5 µg of recombinant SARS-CoV-2-RBD-mFc was labeled and 1.6 µg of labeled protein was used for each replicate. This experiment was conducted in March 2020 as an application of IGC method during the early COVID pandemic. The amount of ligand used for every experiment was added in the revised manuscript.

d. How many cells and how much HGF and/or BSA was used for Fig 2f? etc.

For each group in Fig 2f, 200 ng of HGF was mixed with different amount of BSA (the HGF:BSA ratio was indicated in Fig 2f) and labeled with probe 1. Then, 60 ng of the labeled HGF was used to capture receptors on 2 million HeLa cells per replicate.

e. All bar charts showing MS/MS counts are unclear. Was this MS/MS counts in the crosslinked sample only? How many counts were found in the negative control arm of that experiment?

In all bar charts showing MS/MS counts, all data point for each replicate in both crosslinked and uncrosslinked (negative control) samples were shown.

f. What was the protein concentration in the conditioned media when performing conjugation with probes? Was it left in concentrated form, or was it diluted back to be similar to the conditioned media?

When labeling with the probe, the protein concentration in the conditioned media (CM) was 0.4 µg/µL. The resulting cCM were buffer exchanged by centrifugal ultrafiltration and diluted to the desired volume with PBS. The concentrations of cCM for use were generally higher than the original CM.

g. What concentration was the ligand-probe conjugate solution when incubated with the cells? Was the ligand always diluted to 1 mL no matter how many cells or how much ligand was being used?

The concentration of the ligand-probe conjugate solution depends on both the amount of ligand and the number of cells used, which were indicated in the revised manuscript. The volume of the diluted ligand-probe conjugate solution depends solely on the number of cells. Typically, we diluted the ligand-probe conjugate to 0.2 mL when incubated with 1 million cells to avoid too low concentration or wasting reagents.

5. Why are different negative controls used for nearly every IGC experiment? What effect does the control arm have on the results? Sometimes, glycine is used (Fig 2b). Other times, Tris or BSA is used. In Figure 4d/5c, catalyst +/- catalyst is reported. In Fig 5b, cCM+/cCM- is used, in Supp Fig 10, UV+ vs. UV- is used. What guides this selection of the negative/comparison control? Does using

BSA as a diluent make the results look “nicer” at lower levels than it would with an unrelated negative control (e.g. glycine)? While the different probes may influence some of these choices, the apparently random selection of controls with no discussion around how or why these comparisons were selected, significantly detracts from the usefulness of these approaches for other researchers.

Each IGC experiment is independent and we therefore select the proper controls according to different experimental design. We are sorry for the misunderstanding and added additional annotation as indicated below and in the revised manuscript. We believe that the high selectivity of our approach supports the flexibility for selecting control in related experiments with different designs. To address the reviewer’s concerns and approve our statement, we presented point-by-point explanation and additional experiments as listed below.

For the NHS-based labeling (probe 1), unreacted probe after labeling need to be quenched, and the quench reagent (glycine or Tris) can be used as a negative control. The HATRIC method use glycine as quench reagent and negative control for some experiments. The reason for using glycine as a negative control in Fig. 2b is for comparison with the results of the HATRIC paper [Fig. 2a in *Nat. Commun.* 2018, 9, 1519].

The key criterion is that the negative control does not have the same receptor for the ligand of interest. Using HGF as an example, its receptor MET has no known interaction with insulin (INS), EGF, BSA, Tris or glycine. We have showed that INS and EGF can serve as negative controls in IGC experiments for identification of MET (Fig. 2c and Supplementary Fig. 8c). As shown in Fig. R2, IGC experiments using BSA, Tris, or glycine as negative controls successfully identified MET as the HGF receptor.

Fig. R2 Identification of receptors of HGF (100 ng) from 1 million HeLa cells using the Photo-IGC approach. Equal amounts of (a) BSA, (b) glycine and (c) Tris were used as negative controls, respectively. All experiments were performed in triplicate per condition.

As there are no reports indicating that BSA can specifically bind to any of the cell surface receptors of the ligands used in the manuscript, such as HGF, EGF, PDGFB, SARS-CoV-2 RBD, etc., it is appropriate to use BSA as a negative control. As demonstrated in the additional experiments (Fig. R2a and Fig. R3), and the figures in the manuscript (Fig. 2e and Fig. 6d-e), using BSA as a negative control successfully identified MET and EGFR on human HeLa cells, PDGFRs on mouse NIH 3T3 cells, ACE2 on green monkey Vero E6 cells, and NRP1 on PSCs. In summary, BSA, Tris, and glycine can be interchangeably used as negative controls. Other researchers can also consider using other molecules as negative controls based on their preference. We therefore believe that this versatility does not diminish the usefulness of IGC approaches for other researchers.

Fig. R3 Photo-IGC experiments for the identification of receptors of (a) EGF (100 ng) from 1 million HeLa cells and (b) PDGFB (100 ng) from 1 million NIH 3T3 cells. BSA was used as the negative control. All experiments were performed in triplicate per condition.

When using aminoxy-based probes (probe 2 and 3), unreacted probes and other small molecules were removed by centrifugal ultrafiltration using Amicon Ultra filters (3-kDa cut-off). Glycine or Tris cannot be used as controls since they could also be removed. Therefore, a protein with a larger molecular weight (e.g., BSA in Fig. 3e) can serve as a more suitable negative control than glycine or Tris. It is well known that proteins at low concentrations are easily lost during sample transfer, especially during centrifugal ultrafiltration. The purpose of using BSA as a carrier protein is to avoid the loss of small amounts of ligand (e.g., 200 ng of HGF) during centrifugal ultrafiltration.

The design of crosslinking vs non-crosslinking (i.e., catalyst+/catalyst- and UV+/UV-) experiments is based on the premise that covalently crosslinked receptors can be retained after harsh washing (6M urea and 1% SDS) and thus identified as significantly changed proteins. In the case of IGC experiments with CM (i.e., the whole set of secreted proteins) as ligands, no single protein can serve as a proper negative control. For example, if BSA is used as the control, all biotinylated proteins in CM will be significantly changed proteins. In contrast, the crosslinking/non-crosslinking comparison would reveal receptors rather than ligands since, ideally, the ligands would bind to the cell surface in both experimental and control groups.

The purpose of the cCM+/cCM- comparison was to identify the interacting ligands present in the conditioned media (CM). As mentioned in line 182 of the manuscript, “the interacting secreted proteins were revealed by quantitative proteomic comparison of the cCM-treated and -untreated PSCs (Fig. 5b and Supplementary Fig. 12)”.

6. The data is often presented in a different way for each experiment. For example, in Supp Fig 10 – why is the volcano plot one-sided? Were no proteins found to be higher in the UV- than in the UV+ samples? With such a diverse way of designing and reporting experiments, the potential for these crosslinker designs to be useful for other researchers in other applications is greatly diminished.

We are sorry for this misunderstanding again and acknowledge the reviewer's concerns. In the results of the "all-to-all" IGC experiments, we thought that only proteins that are "upregulated" in the UV+/UV- or catalyst+/catalyst- comparisons are meaningful and interesting to the reader. In an effort to highlight the most relevant receptors or ligands, we chose to display the right side of the volcano plot, especially in the supplementary figures. The corresponding original volcano plots

were provided back in the revised manuscript and Supplementary information.

7. It is somewhat surprising that so few proteins were identified in these experiments (based on the number of dots in the volcano plots). Do you see background binding with Probe 1? How does this compare to Probe 2/3? For each volcano plot, an excel table containing protein information (# peptides identified, quantification in each sample, summary stats, etc) should be provided as supplementary information. As it is, only the few points that the authors chose to label are “identified” for the reader. For example, what are the gray, significantly changing dots in Fig 4d or Supp Fig 10? Why were they identified in the experiment?

We thank for the reviewer to point out the high selectivity of our approach as shown in Fig. b-e. Depending on the scale of the experiment, the background binding with Probe 1 can range from 150-650 protein groups. The number of dots in the volcano plots appear less, as most non-significant dots overlap, and most of the proteins are filtered out by the strict quantification criterion (identified "by MS/MS" in all three replicates in at least one experimental group). It is worth mentioning that the IGC method has lower background binding than conventional pull-down protocols due to the use of harsh washing with a strong buffer containing 6 M urea and 1% SDS. More importantly, a low concentration of ligand-probe conjugates was used for cell surface crosslinking, and most non-bound ligand-probe conjugates were washed away prior to affinity purification by using very limited amount of streptavidin beads. Therefore, the amount of ligand-probe conjugates is much less than the endogenous biotinylated proteins and non-specific binding proteins, and it is expected that the use of different probes would result in similar levels of background binding. For example, in the ligand comparison experiments with HGF-BSA and LIF-BSA mixtures (30 ng:15 μ g) as ligands, all three probes had similar number of non-relevant proteins (all proteins except the ligands and their receptors; Fig. R4).

Fig. R4 The non-relevant proteins (all proteins excluding HGF, LIF, MET, LIFR and IL6ST) identified in IGC experiments with HGF-BSA (30 ng:15 μ g) or LIF-HSA (30 ng:15 μ g) on 1 million HeLa cells. (a) The number of protein groups. (b) The total LFQ intensities of those proteins. All experiments were performed in triplicate per condition.

In fact, Excel tables for each volcano plot containing protein information were provided as the source data in the original submission. We also provided detailed information for each volcano plot, including search results and raw data on ProteomeXchange with the dataset identifier PXD038018 (Reviewer account details: Username: reviewer_pxd038018@ebi.ac.uk; Password: L6YBJgmL).

All quantified proteins were reported, including the gray and significantly changing dots in Fig. 4d or Supplementary Fig. 10. We want to highlight that the data presented in Fig. 4-5 was performed by treating the living cell with probe-conjugated conditional media which containing typically more than 1000 proteins rather than a single protein-of-interest. In this application, unknown ligand-receptor pairs are expected to be identified and appear in the significantly changed region. We therefore only highlighted the well-annotated ligands and receptors.

8.The interaction score was poorly explained. By subtracting the “control” iBAQ value (what samples are the control, anyway??), does this bias for highly expressed proteins?

We apologize for the confusion regarding the explanation of the interaction score. We have modified the experimental design illustration (Fig. 5a) and improved the description of the interaction score for ligands and receptors as follows.

“The iBAQ value reported by MaxQuant for each protein is expected to be proportional to its molar abundance, and the relative molar abundance of each protein can be measured by its relative iBAQ (riBAQ)^{49, 50}. For ligand discovery, the riBAQ values of secreted proteins in secretome data were used as the reference (riBAQ_{ref}). After incubation the labeled secretome (cCM) with cells, the riBAQ values of interacting ligands would be higher than their riBAQ_{ref} since they are captured on cells and the non-interacting proteins are washed away. To remove any possible interfering proteins from the receiving cells, we subtracted the iBAQ value of the ligand in cCM-treated cells from that in CM-untreated cells before calculating riBAQ. Finally, we generated the interaction scores for significant proteins to assess the possibility of a protein acting as an interactor.

$$Interaction\ score = \log_2\left(\frac{riBAQ}{riBAQ_{ref}} + 1\right)$$

To calculate the interaction scores for receptors, we used the riBAQ values of surface proteins in surfaceome data as riBAQ_{ref}. As the receptors that crosslinked with the biotinylated ligands in cCM would be enriched, the riBAQ of receptors in catalyst+ group should be higher than their riBAQ_{ref}. To remove any possible interfering proteins from the cCM (e.g., shed receptors), each significant receptor’s iBAQ value in catalyst+ group was subtracted from that in catalyst– group before calculating riBAQ. An interacting receptor or ligand should have an interaction score greater than 1, with a larger score indicating a greater likelihood of interaction.”

Do the different mass spectrometers used (with different dynamic ranges etc) for the cell surface profiling and the IGC experiments affect the results? Do different MS instruments favor different peptides?

The samples that were quantitatively compared in each experiment were analyzed using the same type of MS instrument to ensure consistency. While most of the IGC experiments using purified ligands were analyzed by Q-Exactive mass spectrometer, more sensitive mass spectrometers were also used for more challenging applications such as the study of PCCs-PSCs interactions. In addition, the application of different type of MS instruments for the same set of experiments should be understandable for project lasting multiple years.

The MS instruments for IGC experiments in Fig. 4 (Q-Exactive) and HeLa surfaceome profiling (Orbitrap Fusion) were different. The purpose of the IGC experiments was to compare the

performance of three probes for identifying receptors on HeLa cells, and the HeLa surfaceome data was used as the same reference to calculate the interaction scores for comparing the three probes (Fig. 4g). Therefore, the change in absolute values of the reference data would not affect the conclusion from the comparison of the three probes. To our knowledge, there are no reports indicating that different MS instruments favor different peptides. However, we recommend to use the same type of mass spectrometer for the reference proteome analysis to reduce the possible inaccuracy of interaction score.

As the real application in Fig. 5, we used the same MS instrument to analyze IGC samples and the reference proteome to calculate interaction scores for discovering functional ligands/receptors. As described in the supplementary information, the samples of IGC experiments on PCC cells and their reference proteome (PCC surfaceome and PSC secretome) were analyzed using a Q Exactive HF-X mass spectrometer. And the samples of IGC experiments on PSC cells and their reference proteome (PSC surfaceome and PCC secretome) were analyzed using a timsTOF Pro mass spectrometer.

9. More discussion (and perhaps data) to compare the three different probes is needed to provide the reader with a comparison of how they work and an indication of how to best choose the probe for each application. When labelling conditioned media proteins/protein mixtures, how efficient is the labelling with Probe 1 and Probe 2? Do they look similar? How do you explain differences between panel a and b in Supp Fig 10? How different are these two probes that should only differ in how they are ligated to their ligands?

We agree with this comment. According to the comment, we added more discussion to compare the three different probes, including the differences in labelling efficiency of Probe 1 and Probe 2 when labelling conditioned media proteins/protein mixtures. The differences between panel a and b in Supp. Fig. 10 was summarized in Fig. 4e and f, and discussed in the revised manuscript. The key information on labeling mechanisms and how to choose the best probe for each application were provided in detail in General Response #2.

10. How many PSC cells were required for all of the experiments described in Figure 5, including surface mapping and PSC conditioned media collection? What is the minimum number of cells required for all-to-all mapping?

For the IGC experiments and surfaceome profiling experiment described in Figure 5, 1 million PSC cells were used for each replicate. The PSC conditioned media were collected from the corresponding cells used in the IGC experiments.

Based on the relatively low amount of quantified PCC surface receptors shown in Fig. 5g, we believe that the amount of secreted proteins collected from 1 million PSC cells was already quite low for all-to-all mapping. Therefore, we did not try a smaller scale experiment.

11. In general the discussion section could be greatly strengthened to comment on the strengths and weaknesses, background binding, etc seen with each of these probes and workflows (what is the role of BSA as a “carrier protein”, how can the interaction score and be used to find new paracrine systems, what are the benefits and drawbacks of each probe? etc.)

As suggested, more discussion of the strengths, weaknesses, and other related aspects of each of these probes and workflows were provided in the revised manuscript.

As explained in the response to question 5, BSA is mainly used as a negative control, and can be serve as a carrier protein for labeling the low-abundance ligands with probe 2/3. The role of carrier proteins is to prevent sample loss of low-abundance proteins, and BSA is a well-established carrier protein (*Commun. Biol.* 2018, 1, 103). The definition of interaction score is also described in the response to question 8.

12. The “double” normalization, first to native biotinylated proteins, then to median, seems questionable. If there are many less proteins in the control, it seems that this second normalization could be quite harsh.

It is also unclear when missing values are being imputed. Earlier, it states that all proteins were identified in all technical replicates before quantification. Why is there then the need to impute values? Did you impute values at the peptide or protein level? Explain why imputation is valid here.

Data normalization to endogenously biotinylated proteins has been reported to reduce variations between biological replicates in streptavidin pull-down (*Anal. Chem.* 2020, 92, 15437). Since each compared sample should contain the same amount of endogenously biotinylated proteins, we applied the normalization to the endogenously biotinylated proteins to normalize sample loadings. We then performed median normalization in an attempt to eliminate systematic bias. Median normalization is one of the widely used normalization techniques that forces the distribution of the log LFQ intensity values to center around median value for each sample (e.g., *Nat. Commun.* 2018, 9, 4230; *Nature* 2020, 587, 657–662; *Nat. Commun.* 2021, 12, 6671; *Nat. Commun.* 2022, 13, 2070; *Nat. Commun.* 2022, 13, 3603).

As described in the “Methods” section, proteins identified in all three replicates in at least one group were selected for quantification. Thus, there are some proteins that only identified in experimental group, but not in control group. For example, in Fig. 2d and R4, pdgfra was identified in all three replicates of the PDGFB group, but it was not found in the control groups. To obtain a complete matrix of LFQ intensities and show those proteins (e.g., pdgfra) in volcano plots, missing value imputation was necessary. Since missing values frequently occur in label-free quantitative proteomics, missing value imputation is a common practice before statistical inference. We imputed the missing values using the default imputation method and parameters in Perseus software, which is one of the most popular tools for MaxQuant outputs analysis (*Nat. Methods* 2016, 13, 731–740).

Fig. R4 The LFQ intensity of the identified pdgfra in Photo-IGC experiment using PDGFB as the ligand (data from Fig. 2d).

13. What was the control in the NRP1 ELISA?

The control in the NRP1 ELISA was the PLAU non-coated well. Accordingly, we changed the annotation “control” to “uncoated” in the figure and the description in the revised manuscript.

Minor comments:

1. Several minor grammatical errors throughout the manuscript that could likely be corrected during proofing stage

Thank you for bringing this to our attention. We have carefully proofread the manuscript and corrected any grammatical errors we found.

2. Full western blots (with full molecular weight range) should be shown uncropped in the supplementary information.

The unprocessed western blot images were provided as Source Data (Source Data Fig. 4h.pdf).

3. Does copper affect cell viability?

Copper toxicity to mammalian cells can be eliminated by the addition of the copper ligand THPTA to form metal-ligand complexes, as reported by Finn's group. (*Bioconjug. Chem.* 2010, 21, 1912; *Angew. Chem. Int. Ed.* 2009, 48, 9879). The Cu^I/THPTA catalyst system has been widely tested in the last decade and is considered safe. Based on Finn's work, which demonstrated that 50 μM Cu (in Cu^I/THPTA system) does not affect cell viability, we used the same catalysis condition for the Click-IGC experiments.

Response to the comments raised by Reviewer #2:

Zheng and colleagues in their article with the title “Deciphering intercellular signalling complexes by interaction-guided chemical proteomics” describe an efficient method how to identify novel ligand-receptor pairs. They utilize three different trifunctional linkers offering certain degree of flexibility how to cross-link the ligand-receptor pair based on their close proximity interaction. Although various chemical proteomic protocols for identification of protein-protein interactions have been developed, the application for analysis of low abundant ligand-receptor interactions is relatively novel. The presented efficiency of the enrichment is impressive and superior to previous publications. However, although the experiments are described and documented well it is not clear what is the main principle(s) which allowed authors to improve the method. The practical efficiency to find new ligand-receptor pairs is relatively limited to one new interaction.

Thank you for the positive and thoughtful comments on our article. We acknowledge that the principles behind the improvement of our method are not fully elucidated, and we have provided a more detailed discussion on the differences between the three different trifunctional probes in the revised manuscript. Although only one new interaction was biologically validated in this study, our

method provided a global all-to-all view of the interacting ligands and receptors in the given cell co-culture system for the first time. We believe that our approach is practical for studying other cell communication systems in an all-to-all fashion.

Major corrections:

The authors must provide a quality control of the MS data. At least, the histograms showing the number of imputed values and Pearson correlations of protein intensities between the replicates. For the main “hits” the supporting information should contain the profile plots, before imputation representing the actual LFQ intensities.

We agree that providing a quality control analysis can help in assessing the quality of the data. We have included histograms that show the number of imputed values, as well as Pearson correlations of protein intensities between the replicates, in the Supplementary information for each figure (Supplementary Fig. 20-104). All IGC experiments in the manuscript achieved good reproducibility for protein quantification (Pearson correlation $r > 0.85$ between three biological replicates). Additionally, profile plots for the main hits identified in the volcano plots are included.

The usage of as low as 1 μL of the streptavidin beads suspension for the enrichment of 10^6 cells is surprisingly low. Majority of the protocols would use $>20 \mu\text{L}$. Is this a crucial aspect of the protocol’s success? How was the beads amount optimized? It might be very challenging to work with such low beads amount, but it might be the critical point for the protocol’s sensitivity.

The amount of beads required for an IGC experiment generally depends on the amount of biotinylated protein in the sample. In contrast to proximity labeling or AP-MS experiments that often involve overexpressed bait proteins, the abundances of labeled or biotinylated proteins in a typical IGC sample are lower than the endogenous biotinylated proteins, such as ACACA, PC, PCCA, MCCC1, and ACACB. For example, in our IGC experiment for identifying receptors of SARS-CoV-2 RBD (Fig. 2e), the LFQ intensity of endogenous biotinylated protein was nearly 100-fold higher than that of the biotinylated ligand/receptor (Fig. R5). Therefore, the amount of beads required can be estimated from the LFQ intensities of captured endogenous biotinylated proteins.

Fig. R5 The LFQ intensity of ACE2, SARS-CoV-2 RBD-mFc, and five endogenous biotinylated proteins in Photo-IGC experiment using SARS-CoV-2 RBD-mFc as the ligand (Fig. 2e).

To determine the optimal amount of streptavidin beads required for the AP-MS analysis, we incubated lysates of 1 million HeLa cells with different volumes of streptavidin beads (1 μ L, 5 μ L, and 10 μ L), followed by AP-MS analysis of the biotinylated proteins. As shown in Fig. R6, we found that 1 μ L of streptavidin beads was sufficient to capture the endogenous biotinylated proteins in 1 million cells. Using an excess amount of beads can lead to more non-specific binding proteins and impair the detection of biotinylated proteins (Fig. R6), even with the harsh wash described in the manuscript. The use of 1 μ L of streptavidin beads per 1 million cells was found to be suitable for all experiments in this manuscript, and we recommend it as a good starting point for other researchers who wish to apply this technique in their experiments. The choice of using 1 μ L of streptavidin beads per 1 million cells was described in line 112 and Supplementary Fig. 8g, h of the revised manuscript.

Fig. R6 The sum of LFQ intensities of the (a) endogenous biotinylated proteins (ACACA, PC, PCCA, MCCC1 and ACACB) and (b) non-biotinylated proteins. All experiments were performed in triplicate per condition.

We conducted the incubation and wash of 1 μ L of streptavidin beads in spin columns (e.g., Pierce #69725) to avoid bead loss. Alternatively, our group has developed a fully integrated spintip-based AP-MS technology, called FISAP, for sensitive interactome profiling (Anal. Chem. 2021, 93, 3026). With the FISAP method, affinity purification with as little as 0.1 μ L of streptavidin beads can be easily performed.

The identification of the novel PLAU receptor NRP1 is certainly of high importance. However, it is not possible to conclude from the provided data, if the NRP1 is actually the major receptor. The direct biochemical comparison of the two receptors is not provided. The enrichment of the ligand-receptor itself (visible in the volcano plot) is always relative depending on the labelling efficiency with the probe, which might be different for the two receptors, degree of the background binding or imputed values. Here, the profile plots before imputation might help to review. The relative higher enrichment of the NRP1 not necessarily reflects the actual ratio on the cells. Although, the acceleration of PSCs proliferation by PLAU is an interesting finding, it is not shown if it is actually linked with the newly discovered NRP1.

We appreciate for the thoughtful comments and agreed that further biochemical experiments are needed to demonstrate that NRP1 is the major receptor responsible for the acceleration of PSCs proliferation by PLAU. As explained in detail in General Response #3, we performed NRP1 knockdown to investigate the role of NRP1, and the results suggest that PLAU-induced cell

proliferation is indeed linked with the newly discovered NRP1.

As suggested by the reviewer, we provided the profile plots (LFQ intensities before normalization and imputation) of NRP1, LRP1 and other significant proteins in Supplementary Fig. 46. Importantly, NRP1 was found to have over 10-fold higher LFQ intensity than LRP1, suggesting NRP1 could play a more important role than LRP1 in PLAU-induced signaling.

It would be beneficial for the field and those applying the presented protocol to have better information how the probes efficiency compares. Is it possible to enrich the PLAU-NRP1 with Probe 1 and 2?

We have added more discussion in the revised manuscript regarding the differences of three probes. Since PLAU is available in purified form, probe 1 is the probe of choice, as demonstrated in Fig. 2. Alternatively, for labeling glycosylated PLAU, probe 2/3 can also be used, as shown in Fig. 3. Although the investigation of PLAU-NRP1 interaction arose from the "all-to-all" IGC experiments conducted with probe 3, the PLAU-NRP1 interaction was verified by the probe 1-based IGC method using purified PLAU as the ligand (Fig. 6d and Fig. 6e).

Fig 4h: The loading control should be added.

The loading control (β -actin) was added in the revised manuscript (Fig. 4h).

Fig 2a: Why the authors did not use as the control not UV irradiated EGF-probe conjugate?

The IGC experiments comparing crosslinking vs. non-crosslinking (i.e., UV+/UV-) can also reveal the receptors of the ligands. As shown in the additional Fig. R7, the Photo-IGC with EGF-probe 1 conjugate as the ligand and UV- as the negative control successfully identified EGFR as the receptor. Theoretically, the UV+/UV- comparison cannot completely remove the bias of non-specific crosslinking when a high concentration of ligand-probe conjugate is used. As we used relatively low amounts of ligands in the manuscript, there is no evidence to suggest that the use of "non-crosslinking" as controls is inappropriate. However, it is more convenient to use unrelated proteins as controls when studying several ligands on the same cells (e.g., Fig. 2c).

Fig. R7 Photo-IGC with 50 ng of EGF on 1 million HeLa cells. The group without UV irradiation was used as a negative control. All experiments were performed in triplicate per condition.

Line 222: In conclusion, it should be clearly stated what are the principal mechanisms behind the method sensitivity and fidelity. The authors should provide the reader with comparison of the probes 1 – 3 to help navigate the selection of the suitable probe for further applications.

We agree that it is important to provide a clear understanding of the mechanisms behind the sensitivity and fidelity of our method. In our revised manuscript, we have added a more detailed discussion on the probes used and their advantages and limitations. Regarding the selection of suitable probes, we have provided a comparison of the three probes used in our study, highlighting the advantages and disadvantages of each. We hope that these revisions will help readers in selecting the appropriate probe for their own studies.

Line 495: It is not possible to access the original data on PRIDE repository. This should be corrected.

We apologize for the data access problem. As the raw dataset is not yet publicly available, it is not accessible by searching the PRIDE Archive or ProteomeXchange. Reviewers will need to log in to PRIDE Archive with the reviewer account provided in the manuscript (Username: reviewer_pxd038018@ebi.ac.uk; Password: L6YBJgmL) and access the data through the “Review Submission” panel. This dataset will be publicly accessible upon publication.

Minor corrections:

Line 102: correct the typos

We have rewritten the sentence in revised manuscript.

Line 208: Please quantify “almost complete silencing”

The sentence was replaced with a quantitative description: “RT-qPCR analysis revealed a more than 70% decrease in PLAU mRNA expression after siRNA treatment (Supplementary Fig. 16), and Western blot analysis also showed a significant decrease in PLAU protein expression (Supplementary Fig. 17).”

Line 362: What is the final concentration of at least on component in click reaction mixture?

In the flow cytometry analysis, we tested three CuAAC catalyst buffers with different concentrations of copper ions (final concentration of 50 μ M, 100 μ M, and 150 μ M), as indicated in Fig. 3c. As suggested, we have added the description of the final concentration of copper ions in the revised manuscript.

Line 373: Similarly, please add the exact final concentration of the protein or probe.

The exact final concentration of the probe and the protein-to-ligand ratio was added in the revised manuscript.

Line 396: What type of the beads were used? Streptavidin-coated magnetic or agarose beads, it would be beneficial to include the product number from the Thermo catalogue.

We apologize for the lack of detail. The beads used were Streptavidin Sepharose beads. We added this information with catalog number in the revised manuscript.

Line 429: two razor and unique peptides were needed to confidently assign the protein. Does it mean that two razor peptides are sufficient as well? Standardly, at least 1 unique peptide should be required for confident identification of the protein. Still, 1 unique peptide might lead to biased quantification.

When MaxQuant is used as the database search platform, “a minimum of two razor and unique peptides” is an acceptable criterion for reporting the number of protein groups identified (e.g., *Nat. Commun.* 2018, 9, 4744; *Nature* 2019, 569, 723; *Nat. Commun.* 2019, 10, 2147; *Nat. Commun.* 2020, 11, 908; *Nat. Commun.* 2021, 12, 3564; *Nat. Commun.* 2022, 13, 3542; *Nat. Commun.* 2023, 14, 472). Since there is no simple filter to eliminate all false positives in a proteomics study, we chose to focus on folding changes and statistical significance rather than on the number of proteins identified. We found that using ≥ 2 razor+ unique peptides was a more stringent criterion than using ≥ 1 unique peptide because it filtered out more unreliable identifications in our proteomics results. In addition, we required proteins to be identified "by MS/MS" in all three replicates in at least one group for protein quantification. After these filtrations, all significant ligands/receptors quantified in the manuscript were manually checked to have ≥ 1 unique peptide. Detailed information for all proteins is provided in the “Source Data”, and the search results and raw data are available for reanalysis.

Response to the comments raised by Reviewer #3:

In this manuscript, the authors develop a trifunctional probe that takes advantage of glycan labeling and click chemistry to probe interactions between secreted ligands and their receptors. They deploy these probes for analysis of single ligands and for all-vs-all analysis of the secretome from one cell type and cell surface receptors from another cell type after crosslinking. The authors identify a new receptor-ligand interaction between NRP1 and PLAU. Although the IGC approach can probe receptor-ligand interactions, it does not seem to directly connect them in a 1-to-1 fashion in the absence of prior knowledge or candidate-based approaches.

We thank the reviewer for precisely summarizing our key methodology novelty. While we agree that our all-vs-all analysis using the IGC approach cannot directly establish receptor-ligand interactions in a 1-to-1 fashion, our method does offer a valuable proteomic technique for generating shortlists of potential ligands and receptors, ranked by the interaction scores. This can be useful for biologists seeking to decipher intercellular signaling on a system scale, rather than in hypothesis-driven manner. It's important to note that any new interactions identified through MS-based proteomics should be validated using additional methods, such as candidate-based approaches.

I was unable to determine, for example, how the authors chose to follow up on the PLAU-NRP1 interaction as it seems there were many other possible pairs in the dataset.

As shown in Fig. 5b, PLAU was found to be the secreted protein with the largest fold change. Furthermore, PLAU was identified as the highest scoring ligand in PCC-to-PSC paracrine signaling (Fig. 5g). Therefore, we decided to further investigate the receptor of PLAU and check whether its receptor was present in the result of all-to-all IGC experiment.

Although the authors claim that HATRIC requires more material, the paper that they cite also includes experiments that use only one million cells, and the cell types used differ, so the experiments are not directly comparable. Both HATRIC and IGC ligands can be synthesized, so their accessibility seems to be the same despite the commercialization of HATRIC. IGC is therefore a potentially complementary technology to existing methods but is unlikely to displace them. The manuscript was a bit difficult to follow and would benefit from substantial revision.

In the HATRIC paper, all the experiments were performed with 20 million cells, and only one case used 1 million cells. In contrast, the IGC method has demonstrated its efficacy with only 1 million cells for both commonly used cancer cell lines and primary cells. Additionally, the example of IGC using only 0.1 million cells should not be ignored, as demonstrated in Fig. 3e.

We have attempted to obtain HATRIC probes for experimental comparison three years ago. We found that the HATRIC and TriCEPS-LRC technology platforms have been commercialized by Dualsystems Biotech AG as proteomics services, but the probes are not for sale. More importantly, the requirement for material (300 µg of ligand and ~60 million cells) contradicts our goal of studying intercellular signaling complexes formed between ligands and receptors at endogenous levels on normal primary cells, as it is too expensive or impossible to obtain such large amounts of normal primary cells and the ligands they secrete.

We agree that the experiments are not directly comparable due to differences in the cell types used. However, several examples in our study (Fig. 2 and Fig. 3) demonstrate that the IGC method requires ~1000-fold less ligands and ~10-fold fewer cells, which is critical for in-situ interrogation of cell-cell communication. Overall, we do not intend to replace the reported methods, but aim to provide the first approach to discover ligand-receptor complexes in cell co-culture system with high sensitivity and selectivity. Accordingly, we have made significantly improvement to our revised manuscript for further clarifying the key novelties.

Response to the comments raised by Reviewer #4:

The work by Zheng and coworkers, represents a smart approach to cover interactions of even lower abundant proteins. Although none of the used functionalities or techniques is new per se, its combination including the synthesis of trifunctional crosslinkers is innovative and allows to profile inter-cell interactions of surface receptors within their native environment – meaning across living cells. The need of led ligands and less cell material is the biggest advantage over other receptor capture techniques and over classical crosslinking mass spectrometry approaches.

On the downside it seems that the workflow shows its full performance only in case of glycosylated

proteins, which limits its applicability.

Overall, the manuscript is well written and will be of value for the community once published. Please find some comments below:

We appreciate for the positive and encouraging comments on our manuscript.

•The authors mention that their spacer arm length of 60 Å is well suited for inter-cell crosslinking. I understand that longer spacer arms will capture more interactors thanks to their extended range and flexibility. However, this might bear the risk to capture of non-specific random interactors that are within the extended linker range within solution and common linker reagents for classical crosslinking mass spectrometry, like, DSS, DSSO, DSBU, etc., are in the range of 10 -20 Å long. Can you comment on these thoughts and especially how you'd ensure to capture only specific interactors?

The key rationale to adopt longer spacer arm is that we specifically aimed to apply them for capturing extracellular glycosylated ligand-receptor complexes with large molecular weight for both the protein complexes and the heavy glycosylation. However, we agree with the reviewer that a shorter spacer arm would likely improve the crosslinking specificity of the probe. In our previous paper (*Proc. Natl. Acad. Sci. U.S.A.* **2018**, *115*, E8863), we compared two trifunctional probes with similar structures and spacer arm lengths of 26 and 56 Å for capturing interacting proteins. We found that the probe with a longer spacer arm performed slightly better in capturing interacting proteins without increasing non-specific interactors. Therefore, we used probes with a spacer arm length of around 60 Å in the manuscript. Our results (Fig. 2b-e, Fig. 3e) also demonstrate the good specificity of these probes. We do not exclude the feasibility of shorter probes, but their crosslinking efficiency should be evaluated beforehand.

•If I understood correctly, in Figure 3C, the authors show by flow cytometry the efficiency of the click chemistry. They nicely show that in the absence of the catalyst Cu(I) no binding takes place as this looks like their control. I was wondering about two things: 1st, what is the control? I guess these are unlabeled cells, but how much Cu(I) was used there? Sorry if I just missed this info.

We apologize for the lack of detail in Figure 3C regarding the efficiency of the click chemistry and the optimal concentration of the catalyst Cu(I). To clarify, the control sample in Figure 3c consisted of cells without Ac₄ManNAz labeling, and no Cu(I) was added. The "no Cu(I)" sample consisted of Ac₄ManNAz labeled cells without Cu(I) addition. The "no Ac₄ManNAz" sample consisted of cells without Ac₄ManNAz labeling, but 50 μM Cu(I) was added. We added the detailed descriptions of each samples in the revised Fig. 3c.

2nd, it's a bit contra-intuitive at first sight that 50 μM Cu(I) showed a better performance than 100 or 150μM. Can you comment on potential reasons? Have you repeated this experiment to validate its outcome? Since the catalyst is not used up during the reaction, I would have assumed the difference between Cu(I) concentrations should be more of kinetic nature, with improved reaction speed the higher the concentration.

We agree with reviewer's view that a higher concentration of Cu(I) should lead to a higher click reaction rate, and that it is counterintuitive that 50 μM Cu(I) showed slightly better performance than 100 or 150 μM . The CuAAC reaction condition for live cell labeling in our study followed the pioneering work by Finn's group (*Bioconjug. Chem.* 2010, 21, 1912; *Angew. Chem. Int. Ed.* 2009, 48, 9879). Finn's work demonstrated that THPTA serves both to accelerate the CuAAC reaction and to protect the cells from damage by oxidative agents produced by the Cu-catalyzed reduction of oxygen by ascorbate. They suggested that Cu concentrations should generally be between 50 and 100 μM , and more than 100 μM Cu is usually not necessary to achieve high rates.

In general, our results are consistent with these previous findings. The difference between 50, 100, and 150 μM Cu might be due to insufficient resuspension of pelleted cells during the click reaction, as some portions of the cells were not biotinylated (the previous Fig. 3c). We repeated this experiment to address the potential issue. As shown in the Fig. R8, 100 and 150 μM Cu(I) showed slightly better labeling than 50 μM Cu(I). The result also confirmed our previous conclusion that "50 μM of Cu(I) catalyst in PBS was sufficient to catalyze the labeling of cells with probe 3 in 15 minutes at 4 $^{\circ}\text{C}$ " (line 127). Since Finn's work have demonstrated that 50 μM Cu (in Cu^I/THPTA system) is a safe concentration to preserve cell viability, we recommend using 50 μM copper for Click-IGC experiments.

Fig. R8 The repeated flow cytometric analysis of Ac₄ManNAz labeled and unlabeled K562 cells incubated with probe 3 in the absence or presence of different concentration of Cu(I), and then conjugated with Streptavidin-Cy3.

General comment:

IGC seems very well suited to detect interaction partners, with higher sensitivity compared to standard affinity purification or other methods as described by the authors. However, since a crosslinker is used, IGC could be even way more powerful in case the actual interaction site could be identified. (At least for future studies) It should be possible to elute the biotin-crosslinked peptides off the streptavidin beads after digestion (e.g. by competition with free biotin, or by introduction of a cleavage group into the linker) and yield a purified sample of the crosslinked peptides for MS analysis. With this the degree of information obtained from IGC experiments would be clearly improved. However, I see that such additional experiments might be more effort than possible in the given time for a revision, especially in case a new adopted probe with an additional cleavage site must be synthesized.

We thank the reviewers for recognizing the good performance of the IGC method. We agree that the method would be more powerful if the crosslinked peptides were identified to provide direct information of protein-protein interaction. The suggested idea of eluting the biotin-crosslinked peptides off the streptavidin beads after digestion is an interesting approach to achieve this goal. In fact, we are working on the development of a new workflow with cleavable trifunctional probes. However, there are two major challenges for MS identification of crosslinked peptides: the efficient fragmentation of large crosslinked peptides and interpretation of their MS2 spectra. As the reviewer kindly suggested, we hope to provide the solution in the near future.

Minor remarks:

- There seem quite some typos or grammar errors distributed over the manuscript. I suggest another proofreading, maybe from a native English speaker. Some that I found:

oLine44: "...that are not necessarily happen in biological microenvironment." Seems grammatically incorrect to me

oLine 66: was/were seems mixed up to me

oLine 102: Seems also grammatically incorrect/not a proper sentence "...since protein is nearly always contains..."

- In Supplemental Figure 18, the figure caption does not fit to the labeling. It seems A) and B) is mixed up.

Thank you for bringing these typos and grammar errors to our attention. All of the mentioned errors have been addressed and corrected in the revised manuscript.

REVIEWERS' COMMENTS

Reviewer #1 (Remarks to the Author):

Most concerns were adequately addressed in this revision with a few remaining issues.

(1) The addition of the siRNA in the proliferation study is a nice addition and shows a potential role for PLAU-NRP1 interaction in proliferation. However, the western blot does not include siRNA knockdown (+) without addition of PLAU (-). Without this control, which was included in the proliferation study, it remains possible that NRP1 knockdown itself lowers/prevents ERK phosphorylation by a PLAU independent mechanism. It also would have been nice to show that the NRP1 knockdown did not affect the levels LRP1 (which the authors also showed is present as an alternative PLAU receptor) as an off-target/knock-on effect, which can be quite common when knocking down cellular receptors (for example, see *J Proteome Res.* 2010 Dec 3; 9(12): 6689–6695, where they show that LRP1 knockdown appears to increase expression of NRP1).

(2) The authors state in their rebuttal that "In all bar charts showing MS/MS counts, all data point for each replicate in both crosslinked and uncrosslinked (negative control) samples were shown". However, I do not see MS/MS counts in the negative control (uncrosslinked) arm of the experiment in Figure 2f, or 3d, or 3f. If zero MS/MS counts were always found in the negative control arm for all of these proteins that should be explicitly stated since the difference in MS counts between control and ligand arms in the experiment could conceivably change with total cell number, etc.

Reviewer #2 (Remarks to the Author):

The authors addressed the points raised in the review. This includes the availability of MS data, quality control of MS analyses and characterisation of NRP1-PLAU interaction. From this perspective, I support the publication. For wide application by the readership, I would consider further rewriting with more focus on a single probe 3, which serves the purpose of all-to-all identification.

Reviewer #3 (Remarks to the Author):

The authors provide substantial new data to follow up on the newly discovered interaction between NRP1 and PLAU. They have also considerably revised the manuscript to provide experimental detail and to make the text easier to follow. My only recommendation is that the authors provide all of the figures in the reviewer response as supplementary information to the paper.

Reviewer #4 (Remarks to the Author):

The authors significantly improved the quality of their manuscript and considered all raised concerns. Usage by others is now facilitated by improvements in their discussion by adding recommendations of potential application for probe 1-3 as well as by adding details to the methods section. Additional support for the novel PLAU – NRP1 crosstalk is given by functional assays using knockdowns. If this reflects a direct physical interaction (or an indirect connection including other partners) is however not confirmed by methods other than the here reported IGC. Confirmation could be done using complementary biochemical approaches as a co-immunoprecipitation + western blot or by AP-MS.

Although additional to understand the PLAU – NRP1 crosstalk more in detail would be very valuable, considering the technological/methodological focus of this work, I give an overall recommendation for publication. I believe the IGC approach will be of value for the community, especially in case the used probes are commercialized in future, enabling easy access.

Response to the comments raised by Reviewer 1:

Most concerns were adequately addressed in this revision with a few remaining issues.

(1) The addition of the siRNA in the proliferation study is a nice addition and shows a potential role for PLAU-NRP1 interaction in proliferation. However, the western blot does not include siRNA knockdown (+) without addition of PLAU (-). Without this control, which was included in the proliferation study, it remains possible that NRP1 knockdown itself lowers/prevents ERK phosphorylation by a PLAU independent mechanism.

Thank you for your insightful comments on our revised manuscript. We appreciate your thorough evaluation of our work. We have carefully considered your suggestions and have addressed most of the concerns raised in this revision. We would like to respond to the remaining issues you have highlighted.

Regarding your first point, we acknowledge the importance of including a control to demonstrate that NRP1 knockdown itself does not affect ERK phosphorylation by a PLAU-independent mechanism. We performed additional experiments to include the siRNA knockdown (+) without the addition of PLAU (-) control (**Fig. R1**). We have now included this control in the revised manuscript,

and the corresponding results are presented in Figure 6g of the revised manuscript.

Fig. R1 Western blot assay of NRP1, LRP1, pERK and ERK in PSC cells transfected with siControl or NRP1 targeting siRNAs (siNRP1) and treated with PLAU at 100 ng/mL for 72h. Images are representative of 3 biological replicates.

It also would have been nice to show that the NRP1 knockdown did not affect the levels LRP1 (which the authors also showed is present as an alternative PLAU receptor) as an off-target/knock-on effect, which can be quite common when knocking down cellular receptors (for example, see *J Proteome Res.* 2010 Dec 3; 9(12): 6689–6695, where they show that LRP1 knockdown appears to increase expression of NRP1).

We appreciate your suggestion to assess the effect of NRP1 knockdown on the levels of LRP1 as an off-target or knock-on effect. We agree that such an assessment is important to exclude any potential confounding effects of NRP1 knockdown on LRP1 expression. To address this concern, we performed additional experiments to examine the levels of LRP1 upon NRP1 knockdown. The results confirm that NRP1 knockdown does not affect the expression of LRP1, as demonstrated by western blot analysis (**Fig. R1**).

(2) The authors state in their rebuttal that “In all bar charts showing MS/MS counts, all data point for each replicate in both crosslinked and uncrosslinked (negative control) samples were shown”. However, I do not see MS/MS counts in the negative control (uncrosslinked) arm of the experiment in Figure 2f, or 3d, or 3f. If zero MS/MS counts were always found in the negative control arm for all of these proteins that should be explicitly stated since the difference in MS counts between control and ligand arms in the experiment could conceivably change with total cell number, etc.

We apologize for our misunderstanding and the inaccurate description in the previous response: “In all bar charts showing MS/MS counts, all data point for each replicate in both crosslinked and uncrosslinked (negative control) samples were shown”. In the experiment in Figure 2f, or 3d, or 3f, we did not use uncrosslinked samples as negative control. In the experiment shown in Figure 2f, each condition can be compared with one another since the only differing factor between the compared conditions is the ligand/protein used, while other variables, including cell number, remain constant. We have demonstrated that Photo-IGC with HGF as ligand specifically identified MET as

receptor (Fig. 2c and Supplementary, Fig. 8a-d), while the BSA control does not capture MET (Supplementary, Fig. 8c). Therefore, the MS/MS counts of MET were used as an indicator to compare the performance between the different HGF:BSA ratios shown in Figure 2f.

Similarly, in Figure 3d, the only factor that differs between the compared conditions is the scale (cell number) of experiment. The aim of this experiment was to find the minimal number of cells required for Click-IGC method, and the finding that 0.1 million cells were sufficient for Click-IGC experiment was further demonstrated in Fig. 3e. As for the experiments in Figure 3f, the two protein mixtures were compared with each other as a control for each probe.

In summary, we consider those bar charts (Figure 2f, or 3d, or 3f) were sufficient to support the corresponding conclusions in the manuscript without adding the uncrosslinked samples as additional controls.

Thank you once again for your valuable input, which has significantly improved the quality and rigor of our study. We hope that these revisions address the remaining concerns and strengthen the scientific integrity of our study.

Response to the comments raised by Reviewer #2:

The authors addressed the points raised in the review. This includes the availability of MS data, quality control of MS analyses and characterisation of NRP1-PLAU interaction. From this perspective, I support the publication. For wide application by the readership, I would consider further rewriting with more focus on a single probe 3, which serves the purpose of all-to-all identification.

We appreciate your positive feedback regarding our efforts to address the points raised in the previous review. Furthermore, we appreciate your recognition of the performance of Probe 3 and your suggestion regarding the rewriting of the manuscript to provide more focus on probe 3. However, probe 1 and probe 2 are important alternative solutions when metabolic labeling is not applicable, as described in the Discussion section. Additionally, we also utilize probe 1 to validate the PLAU receptor due to its selectivity and ease of use. Therefore, we would like to retain the content of the Probe 1/2-based Photo-IGC in the main text. By discussing multiple probes, we can provide a broader understanding of the experimental design, the rationale behind probe selection, and the potential strengths and limitations of each probe. Regarding your suggestion, we also modified the Discussion section to give appropriate emphasis to probe 3.

Response to the comments raised by Reviewer #3:

The authors provide substantial new data to follow up on the newly discovered interaction between NRP1 and PLAU. They have also considerably revised the manuscript to provide experimental detail and to make the text easier to follow. My only recommendation is that the authors provide all of the figures in the reviewer response as supplementary information to the paper.

Thank you for your positive evaluation of our revised manuscript. We appreciate your

recommendation to provide all the figures mentioned in the reviewer response as supplementary information to the paper. All figures from the previous reviewer responses (Figures R1, R2, R3, R4, R5, R6, R7, and R8) have been added to the revised Supplementary Information.

Response to the comments raised by Reviewer #4:

The authors significantly improved the quality of their manuscript and considered all raised concerns. Usage by others is now facilitated by improvements in their discussion by adding recommendations of potential application for probe 1-3 as well as by adding details to the methods section. Additional support for the novel PLAU – NRP1 crosstalk is given by functional assays using knockdowns. If this reflects a direct physical interaction (or an indirect connection including other partners) is however not confirmed by methods other than the here reported IGC. Confirmation could be done using complementary biochemical approaches as a co-immunoprecipitation + western blot or by AP-MS.

Although additional to understand the PLAU – NRP1 crosstalk more in detail would be very valuable, considering the technological/methodological focus of this work, I give an overall recommendation for publication. I believe the IGC approach will be of value for the community, especially in case the used probes are commercialized in future, enabling easy access.

Thank you for your positive feedback and acknowledgment of the improvements we made to address the raised concerns. We are grateful for your suggestions regarding complementary biochemical approaches such as co-immunoprecipitation or AP-MS, which could be employed in future investigations. We appreciate your recognition of the technological/methodological focus of our work and your overall recommendation for publication. We also share your belief that if the used probes are commercialized in the future, it will enable easy access and further benefit the research community.